# Integrative structure determination reveals functional global flexibility for an ultra-multimodular arabinanase

Shifra Lansky [1][✉], Rachel Salama[2], Xevi Biarnés [3], Omer Shwartstein[1], Dina Schneidman-Duhovny[4], Antoni Planas [3], Yuval Shoham[2][✉] & Gil Shoham [1][✉]

AbnA is an extracellular GH43 α-L-arabinanase from *Geobacillus stearothermophilus*, a key bacterial enzyme in the degradation and utilization of arabinan. We present herein its full-length crystal structure, revealing the only ultra-multimodular architecture and the largest structure to be reported so far within the GH43 family. Additionally, the structure of AbnA appears to contain two domains belonging to new uncharacterized carbohydrate-binding module (CBM) families. Three crystallographic conformational states are determined for AbnA, and this conformational flexibility is thoroughly investigated further using the "integrative structure determination" approach, integrating molecular dynamics, metadynamics, normal mode analysis, small angle X-ray scattering, dynamic light scattering, cross-linking, and kinetic experiments to reveal large functional conformational changes for AbnA, involving up to ~100 Å movement in the relative positions of its domains. The integrative structure determination approach demonstrated here may apply also to the conformational study of other ultra-multimodular proteins of diverse functions and structures.

[1] Institute of Chemistry, the Hebrew University of Jerusalem, Jerusalem 91904, Israel. [2] Department of Biotechnology and Food Engineering, Technion, Haifa 3200, Israel. [3] Laboratory of Biochemistry, Institut Químic de Sarrià, Universitat Ramon Llull, Barcelona 08017, Spain. [4] School of Computer Science and Engineering, the Hebrew University of Jerusalem, Jerusalem 91904, Israel. [✉]email: shifra.lansky@mail.huji.ac.il; yshoham@bfe.technion.ac.il; gil2@vms.huji.ac.il

Carbohydrate active enzymes (CAZymes)[1,2] are a widespread group of enzymes with numerous substrate specificities suitable for the extreme diversity of carbohydrates. At least in terms of quantities, these enzymes are the most abundant in nature, since about two thirds of the carbon in the biosphere is in the form of carbohydrates, mostly as cellulose and hemicellulose. The growing interest in CAZymes stems from their wide usage potential and especially from their two main applications: they are pivotal for the generation of second generation bioethanol from lignocellulose (biomass)[3,4] and they can be used for the synthesis of complex carbohydrate containing compounds[5–7].

CAZymes include glycosyl hydrolases (GHs) (EC 3.2.1.-)[8–14], which hydrolyze glycosidic bonds; glycosyl transferases (GTs) (EC 2.4.1.-)[15], which form new glycosidic bonds; carbohydrate esterases (CEs) EC 3.1.1.-)[16–18], which hydrolyze the ester bonds of carbohydrate sidechains; polysaccharide lyases (PLs) (EC 4.2.2.-)[19], which cleave glycosidc bonds through non-hydrolytic mechanisms; and enzymes of auxiliary activities (AAs) (1.14.99.-)[20], specifically, lytic polysaccharide mono-oxygenases (LPMOs), which cleave carbohydrates through redox reactions[21,22]. A most informative and updated classification of this wide range of carbohydrate active enzymes is available in the Carbohydrate-Active Enzymes database (CAZy; http://www.cazy.org)[1].

CAZymes often contain auxiliary domains in addition to their catalytic domains, or even more than one catalytic domain[23], generating large, multimodular and multifunctional assemblies. Such assemblies assist the enzymes to degrade complex, recalcitrant, and solid substrates, and to display an overall improved catalytic efficiency[24]. Ultra-multimodular CAZymes are considered as enzymes that have three or more independently folded modules[25], however, despite their very large number in the CAZy database, a relatively small number of full-length experimental structures[26–29] and conformational studies[30–33] are available for them. This is owing mainly to their high flexibility and conformational heterogeneity, which impedes their structural study.

AbnA is an extracellular arabinanase from the Gram-positive thermophilic bacterium *Geobacillus stearothermophilus*[34], belonging to the glycosyl hydrolase family GH43. AbnA is capable of degrading long polymeric arabinan chains, and is a key enzyme in the degradation and utilization of the insoluble polysaccharide arabinan[34]. Besides the application of arabinanases such as AbnA in polysaccharide degradation for the generation of renewable energy, these enzymes have notable roles in nutritional medical research, food industry, and organic synthesis[35]. Structures of homologous enzymes from the GH43 family have been solved in the past[36,37], however, they are much smaller than AbnA, covering only ~60% of the AbnA sequence. AbnA possess 848 amino-acids, and is predicted to possess an ultra-multimodular assembly. Thus, besides the high biotechnological potential, the interest in the structural-function analysis of AbnA stems also from its large size and its possession of additional domains that have not yet been characterized within the GH43 family.

In the present work, we report X-ray crystal structures of AbnA in various conformations, displaying a multi-domain "pincer-like" structure. The structure of AbnA is the largest GH43 structure to be determined so far, and the only ultra-multimodular assembly to be solved within this family. Additionally, it appears to contain two domains belonging to new uncharacterized carbohydrate-binding module (CBM) families. Molecular dynamics (MD), metadynamics, normal mode analysis (NMA), small angle X-ray scattering (SAXS), dynamic light scattering (DLS), cross-linking, and kinetic experiments have further been used to investigate additional conformational states for AbnA, suggesting that AbnA undergoes large functional conformational changes involving up to ~100 Å movement in the relative positions of its domains. In this study, through the combination of multiple complementary methods, both experimental and computational, we demonstrate the use of the integrative structure determination approach to study the functional global flexibility of AbnA, an approach that may also apply to the study of many other ultra-multimodular enzymes.

## Results

**X-ray crystallography data reveals the structure of AbnA.** X-ray crystallography data were collected on an AbnA-WT crystal, leading to its structure determination at 2.35 Å resolution (Table 1). The structure of AbnA is the largest structure to be solved within the GH43 family (848 amino acids; 94 kDa), and contains four domains instead of the one or two domains usually found in enzymes of this family (Fig. 1). The four domains of AbnA are composed of residues 1–441 (**Domain1**), 442–550 (**Domain2**), 551–639 (**Domain3**), and 640–848 (**Domain4**), arranged in an overall "pincer-like" structure. Domain1 and Domain2 are connected tightly one to the other, with 2162 Å² surface area buried in between them. This contrasts with the relative independence of Domain3 and Domain4, which are connected to the other domains solely by single loops.

Domain1, the catalytic domain of AbnA, corresponds to the typical five-bladed β-propeller fold observed in GH43 enzymes. This fold consists primarily of five radial β-sheets, built each of three to four β-strands. The active site of AbnA is located in the middle of a long binding cleft at the center of Domain1. The catalytic residues, all situated on the blades of the propeller, consist of the general base Asp55, the general acid Glu304, the pKa modulator Asp238, and His395, which coordinates a calcium ion at the center of the enzyme. Domain2 interacts tightly with Domain1, and resembles the extra domains found in the structures of other GH43-subfamily-4 enzymes[36,37], adopting a distorted β-barrel fold comprised of eight central β-sheets (Fig. 1a).

Domain3 and Domain4 of AbnA present the first examples of such domains seen so far in arabinanases of the GH43 enzymes. Domain3 is composed of three long anti-parallel β-strands and a short α-helix. This domain links Domain2 and Domain4 together, and as discussed further below, seems to function as a "hinge" domain around which the other domains rotate. Domain4 is a two-layered β-sandwich domain, built of a convex layer of nine anti-parallel β-strands, and a concave layer of seven anti-parallel β-strands containing many Trp residues. A calcium ion, octahedrally-coordinated, was modeled in the convex layer (Fig. 1a). The general fold of this domain bears resemblance to the concanavalin A-like lectin/glucanase fold, and more specifically to the fold often observed in glycan chain binding (type B) carbohydrate-binding modules (CBMs) belonging to the β-sandwich superfamily[38], specifically in the CBM29[39,40], CBM22[41], and CBM4[42] families. This similarity in fold suggests that Domain4 of AbnA functions as a CBM domain. However, because Domain4 differs considerably in the number of β-strands in each sheet and does not contain sequence identity to these 3 specific CBM families, it appears that Domain4 of AbnA represents a CBM domain belonging to a new CBM family, not yet characterized.

**AbnA possesses two substrate-binding sites.** To capture the substrate in the active site, an AbnA crystal was soaked in an arabinopentaose (A5) solution prior flash-freezing, producing a new AbnA structure at 2.95 Å resolution. Although an arabino-oligosaccharide molecule was not observed in the active site, extra density was observed in Domain4, and this could be clearly modelled as an arabinopentaose molecule (**AbnA-Conf1-A5**). The trapped arabinopentaose is situated in the concave cleft of Domain4, amongst the many aromatic residues present in the region, right in front of the active site of Domain1 (Fig. 1b, d,

**Table 1 Data collection and refinement statistics of the AbnA structures.**

| | AbnA-Conf1 | AbnA-Conf2 | AbnA-Conf3 | AbnA-A5 | AbnA-D123-A8 |
|---|---|---|---|---|---|
| *Data collection* | | | | | |
| Space group | $P2_12_12_1$ | $P2_12_12_1$ | $P2_12_12_1$ | $P2_12_12_1$ | $P6_122$ |
| Cell dimensions | | | | | |
| $a, b, c$ (Å) | 69.7, 87.5, 129.2 | 73.7, 82.6, 131.7 | 72.8, 82.6, 132.1 | 70.4, 87.8, 129.9 | 129.2, 129.2, 488.4 |
| $\alpha, \beta, \gamma$ (°) | 90, 90, 90 | 90, 90, 90 | 90, 90, 90 | 90, 90, 90 | 90, 90, 120 |
| Resolution (Å) | 50.00–2.36 | 50.00–2.35 | 48.93–2.09 | 50.00–2.95 | 48.84–2.84 |
| | (2.40–2.36)[a] | (2.39–2.35) | (2.19–2.09) | (3.00–2.95) | (3.02–2.84) |
| $R_{sym}$ or $R_{merge}$ | 8.8 (40.3)[a] | 10.4 (56.0) | 7.1 (52.9) | 21.6 (72.0) | 12.7 (63.3) |
| $I / \sigma I$ | 10.4 (3.4)[a] | 8.4 (2.2) | 23.2 (3.6) | 5.1 (3.6) | 35.6 (7.2) |
| Completeness (%) | 99.1 (96.5)[a] | 98.0 (86.0) | 97.6 (86.0) | 100.0 (100.0) | 99.9 (99.5) |
| Redundancy | 4.3 (4.0)[a] | 6.0 (4.5) | 11.34 (6.2) | 12.4 (10.5) | 37.6 (39.1) |
| *Refinement* | | | | | |
| Resolution (Å) | 50.00-**2.36** | 50.00-**2.35** | 48.93-**2.09** | 50.00-**2.95** | 48.84-**2.84** |
| No. reflections | 30,639 | 31,763 | 43,213 | 16,471 | 55,642 |
| $R_{work}/R_{free}$ | **16.5**/24.2 | **17.8**/25.5 | **19.2**/25.2 | **16.8**/25.9 | **18.3**/24.2 |
| No. atoms | | | | | |
| Protein | 6353 | 6308 | 6316 | 6359 | 9408 |
| Ligand/ion | 28 | 30 | 18 | 68 | 308 |
| Water | 378 | 427 | 275 | 142 | 157 |
| *B*-factors | | | | | |
| Protein | 27.9 | 36.9 | 45.8 | 33.4 | 45.7 |
| Ligand/ion | 35.2 | 51.1 | 44.6 | 47.5 | 80.7 |
| Water | 27.8 | 33.1 | 43.2 | 20.1 | 41.6 |
| R.m.s. deviations | | | | | |
| Bond lengths (Å) | 0.014 | 0.013 | 0.009 | 0.013 | 0.012 |
| Bond angles (°) | 1.60 | 1.56 | 1.16 | 1.60 | 2.29 |

[a]Values in parentheses are for highest-resolution shell.

Supplementary Fig. 1a). These residues bind the substrate through many π−interactions, resembling the binding mode commonly observed in CBMs of the β-sandwich superfamily[38].

To confirm the biochemical relevance of the Domain4 binding site, isothermal calorimetry (ITC) was used to measure the heat released upon binding of linear and branched arabinan to the nucleophile catalytic mutant (E304A) of AbnA and different truncation forms of AbnA (AbnA-D123, AbnA-D12, AbnA-D4, AbnA-D34) (Table 2, Supplementary Fig. 2). The binding interactions were exothermic and enthalpically-driven in all cases, with enthalpy values ranging between −8.6 and −31.1 kcal mol⁻¹ and $K_d$ constants between 1.4 and 11.1 μM. Thus, both Domain1 and Domain4 were shown to bind arabinan independently of each other, confirming the biochemical relevance of both of the substrate binding sites of AbnA.

The capture of an arabino-oligosaccharide substrate in the catalytic active site of AbnA finally succeeded when co-crystallizing arabinooctaose (A8) with a truncated form of AbnA lacking Domain4 (**AbnA-D123-A8**). Structure determination revealed high similarity to Domain1, Domain2 and Domain3 of AbnA-WT (RMSD of 0.85 Å), and an arabinooctaose molecule trapped in the active site, modeled at ~75% occupancy. The trapped arabinooctaose spans the full length of the binding cleft in Domain1, enabling complete mapping of the eight sugar binding sub-sites (−4 to +4) (Fig. 1c, Supplementary Fig. 1b). Interestingly, when superposing the AbnA-D123-A8 structure upon the AbnA-Conf1-A5 structure, it is apparent that the arabino-oligosaccharide bound to Domain1 is in a perpendicular orientation to the arabino-oligosaccharide bound to Domain4 (Fig. 1d).

**Different crystallographic conformational states determined for AbnA**. A second crystallographic dataset was obtained for AbnA-WT under slightly different cryogenic conditions (10% ethylene glycol instead of 10% PEG400). The corresponding

structure determination revealed a more compact ("closed") conformational state of the protein, showing significant interactions between Domain1 and Domain4. The original conformation (**AbnA-Conf1**) and the new conformation (**AbnA-Conf2**) differ primarily by a 10–13 Å global movement in the positions of Domain3 and Domain4 relative to Domain1 and Domain2, closing the open pincer structure of the protein observed in AbnA-Conf1 (Fig. 2a, b). In the AbnA-Conf2 structure, three new hydrogen bonds mediate the interaction between Domain1 and Domain4, specifically the pairs Glu420-Trp758, Tyr421-Ser756, and Glu420-Ser755 (Fig. 2b).

A third global conformational state was observed for AbnA following soaking experiments with an arabinoheptaose substrate. These experiments did not result in a bound substrate in the active site; however, they revealed another conformational state of the protein. This state, termed **AbnA-Conf3**, resembles the AbnA-Conf2 conformation in regards to the interactions that take place between Domain1 and Domain4. However, superposition of these two structures demonstrates that in the AbnA-Conf3 structure, Domain4 is rotated around itself by approximately 45° relative to its position in the AbnA-Conf2 structure (Fig. 2c). Similarly to the AbnA-Conf2 structure, also in the AbnA-Conf3 structure, hydrogen bonds mediate the interaction between Domain1 and Domain4, specifically a hydrogen bond between residues Asp329 and His736, and a water-mediated hydrogen bond between Glu420 and Asn735 (Fig. 2d).

**NMA sampling supports the crystallographic conformational states**. To support the biochemical relevance and significance of the three different crystallographic conformational states, normal mode analysis (NMA) was performed. NMA predicts and analyzes large-scale collective motions in proteins based on calculation of their low-frequency modes[43]. The structure of AbnA-Conf2 was submitted to the NOMAD-Ref webserver[44], which

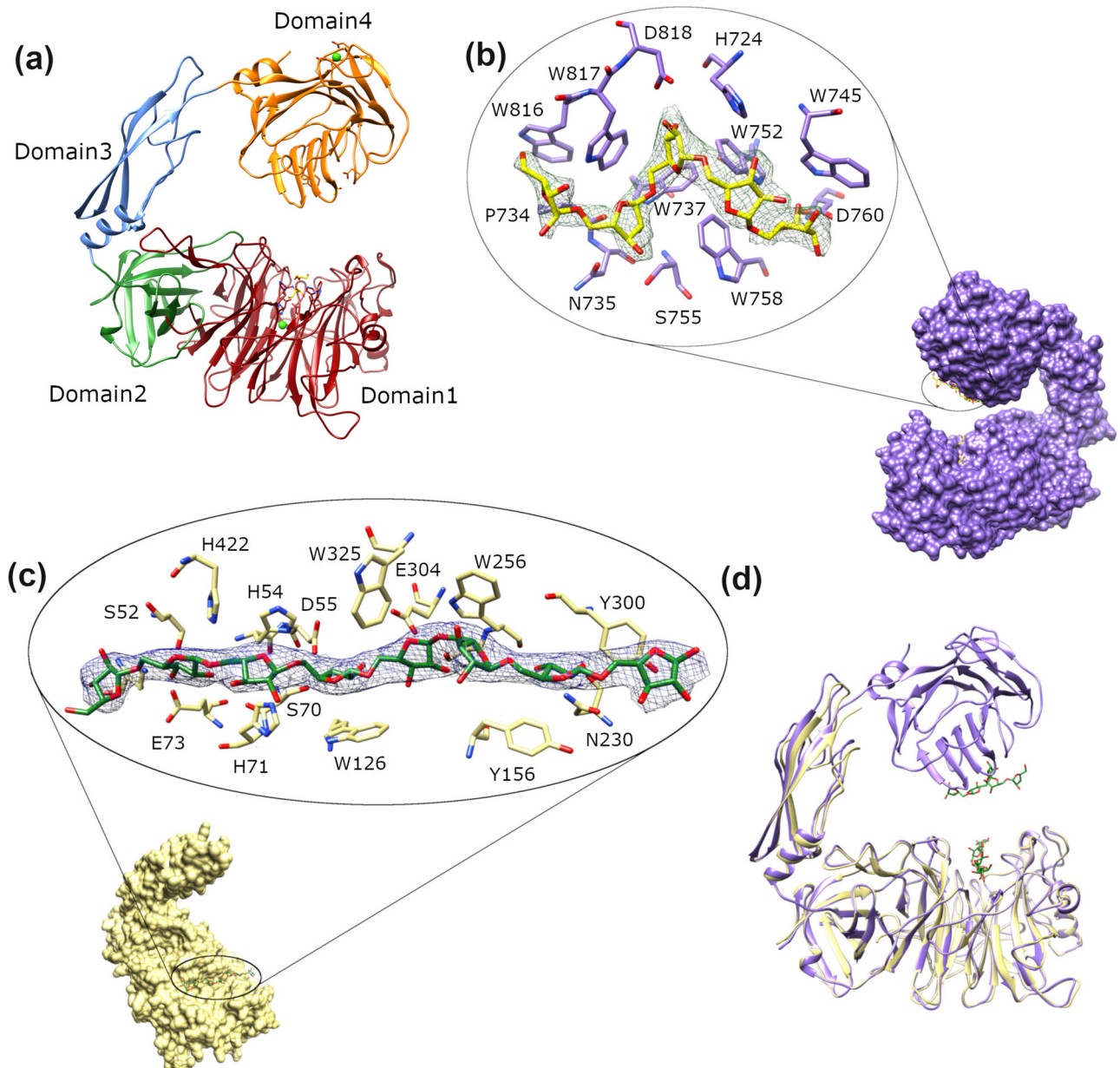

**Fig. 1 The crystal structure of AbnA and its complexes. a** The pincer-shaped structure of AbnA-Conf1, composed of the catalytic Domain1 (in red), Domain2 (green), Domain3 (blue), and Domain4 (orange). **b** The structure of AbnA-Conf1-A5, where an arabinopentaose (A5) molecule was found bound to Domain4. **c** The structure of the AbnA-D123 truncation mutant in complex with an arabinooctaose (A8) molecule in the active site. **d** Superposition of the AbnA-Conf1-A5 structure (purple) with the AbnA-D123-A8 structure (yellow), revealing that the arabinosaccharides present in the two binding sites of AbnA are bound in a perpendicular orientation to one another.

provides tools for online calculation of the normal modes of large molecules based on Cartesian coordinate space. The lowest frequency mode that was calculated corresponded to a "sideways" motion of Domain4 relative to the other domains (Fig. 3a), analogous to the changes observed between the AbnA-Conf2 and AbnA-Conf3 structures (Fig. 2c). The next low frequency mode corresponded to an "opening-up" motion of the AbnA pincer (Fig. 3b), analogous to the changes observed between the AbnA-Conf2 and AbnA-Conf1 structures (Fig. 2a). NMA calculations were also done with the iModS normal mode analysis webserver[45], which calculates these modes in dihedral space (according to torsion angles), facilitating preservation of correct geometry and enabling faster calculations. This server predicted for the lowest-frequency mode a combination of the two modes obtained with NOMAD-Ref, i.e., a "sideways" and an "opening-

up" motion of Domain4 (Fig. 3c, d). Thus, NMA results reinforce the biochemical relevance and significance of the three observed crystallographic conformational states, thereby suggesting the possibility of additional states.

**Additional conformational states sampled and scored by MD and metadynamics.** To investigate further the possibility of additional conformational states in solution, a 500 ns all-atom classical molecular dynamics (MD) simulation was performed, starting from the AbnA-Conf2 crystal structure of AbnA. The time trajectory that was obtained was then analyzed with respect to the distance between Domain1 and Domain4 (representing an "opening-up" motion) and the dihedral angle between Domain1, Domain2, Domain3, and Domain4 (representing a "sideways" motion). Such

**Table 2 Thermodynamic parameters of the binding of AbnA and its variants to branched and linear arabinan.**

| AbnA variant | $K_B \times 10^5$ (M$^{-1}$) | $K_D$ $^{(1/K_B)}$ (µM) | $\Delta H_B$ [kcal/mol] | $T\Delta S_B$ [kcal/mol] | $\Delta G_B$ [kcal/mol] | Calculated effective binding units of arabinose[a] |
|---|---|---|---|---|---|---|
| *Sugar-beet arabinan* | | | | | | |
| Domains1234 | 2.3 ± 0.3 | 4.3 ± 0.6 | −26.1 ± 0.5 | −18.7 ± 0.5 | −7.43 ± 0.08 | 25 |
| Domains123 | 1.8 ± 0.3 | 5.5 ± 0.9 | −8.6 ± 0.2 | −1.3 ± 0.2 | −7.28 ± 0.10 | 14 |
| Domains12 | 3.6 ± 0.6 | 2.8 ± 0.5 | −15.3 ± 0.2 | −7.6 ± 0.2 | −7.69 ± 0.10 | 23 |
| Domain4 | 1.3 ± 0.1 | 7.7 ± 0.6 | −17.7 ± 0.2 | −10.6 ± 0.2 | −7.08 ± 0.05 | 11 |
| Domains34 | 1.6 ± 0.1 | 6.2 ± 0.4 | −17.5 ± 0.2 | −10.3 ± 0.2 | −7.21 ± 0.04 | 13 |
| *Linear arabinan* | | | | | | |
| Domains1234 | 3.0 ± 0.4 | 3.3 ± 0.4 | −31.1 ± 0.5 | −23.5 ± 0.5 | −7.59 ± 0.08 | 22 |
| Domains123 | 7.1 ± 1.2 | 1.4 ± 0.2 | −14.8 ± 0.2 | −6.7 ± 0.2 | −8.11 ± 0.10 | 13 |
| Domains12 | 4.5 ± 0.6 | 2.2 ± 0.3 | −13.5 ± 0.2 | −5.7 ± 0.2 | −7.83 ± 0.08 | 15 |
| Domain4 | 0.9 ± 0.1 | 11.1 ± 1.2 | −16.7 ± 0.3 | −9.8 ± 0.3 | −6.86 ± 0.07 | 11 |

[a]To calculate the binding constants, the molar ratio was fixed to 1, and the size of the "effective binding unit" for the polymer was calculated. The effective binding unit represents how many monosaccharide equivalents of polysaccharide chain is, on average, are required for binding (or are masked) by the protein molecule. In all the variants that include Domain1, the acid base mutant E304A was used.

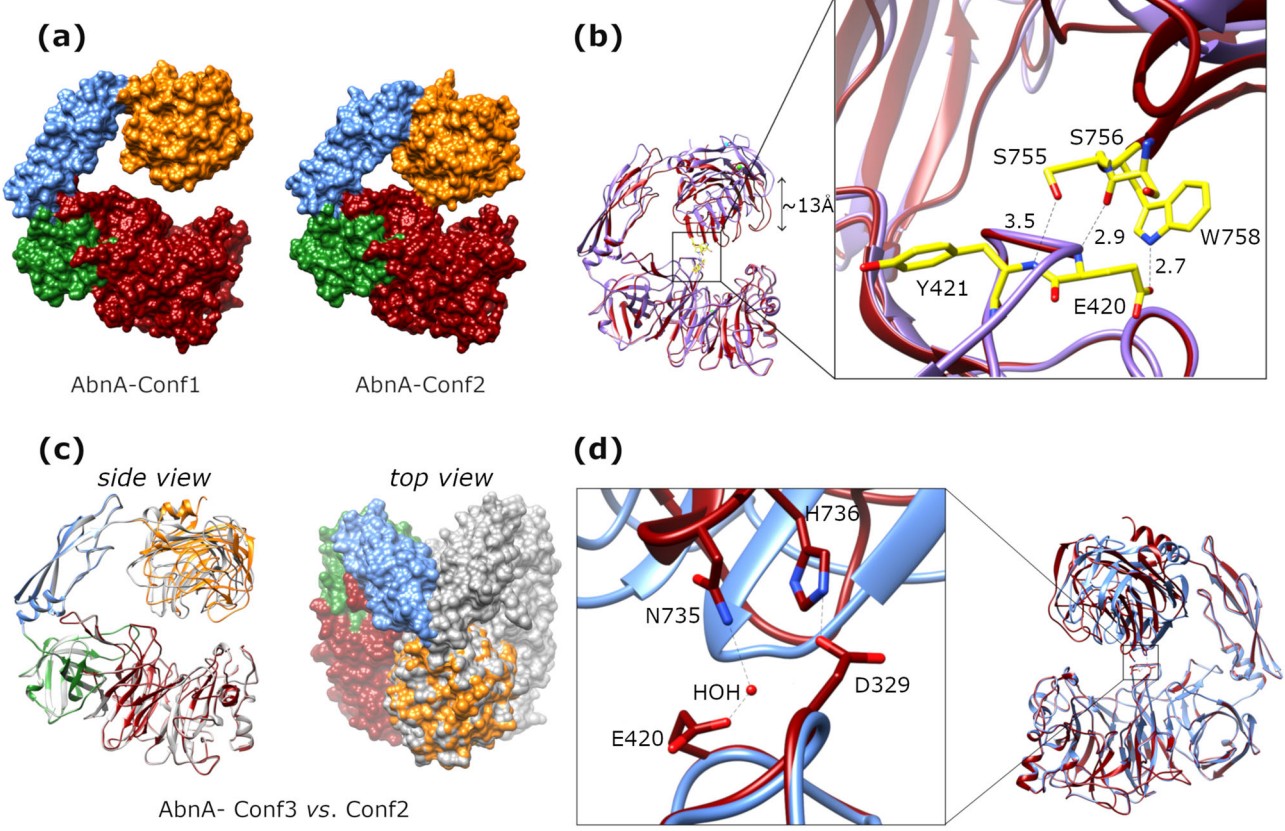

**Fig. 2 The three crystal conformations determined for AbnA. a** Comparison between the AbnA-Conf1 and the AbnA-Conf2 structures, revealing an opening-closing movement of Domain4. **b** Superposition of the AbnA-Conf1 (in purple) and AbnA-Conf2 (in red) structures, revealing a ~13 Å downward movement in the relative position of Domain4. This comparison shows that only in the AbnA-Conf2 structure, unlike the AbnA-Conf1 structure, there are hydrogen bond interactions between Domain1 and Domain4. **c** Superposition of the AbnA-Conf3 (in color) and the AbnA-conf2 (in gray) structures one on the other, viewed from the side (alignment based on Domain 1) and from above (alignment based on Domain4). Such superposition reveals an internal rotation of Domain4 by about 45° relative to the AbnA-Conf2 structure. **d** Superposition of the AbnA-Conf2 (in blue) and AbnA-Conf3 (in red) structures reveals that the two structures possess different interactions between Domain1 and Domain4.

analysis revealed that the crystal structure opened widely to a distance of ~70 Å between Domain1 and Domain4 after 50 ns, opened to a lesser extent after 100 ns and 130 ns, but then remained closed for the remaining duration of the simulation (Fig. 4a).

To examine further the wide-open conformations that were observed during the AbnA-Conf2 classical MD simulation (conformations that were relatively rarely sampled), a metadynamics

simulation was performed. This was done to enhance the sampling of rare conformations and to map the free energy landscape of the different possible conformations. In this simulation, two collective variables (CVs) were used to bias the energy potential, the distance between the centers of mass of Domain1 and Domain4 (CV1), and the dihedral angle between the centers of mass of Domain1, Domain2, Domain3, and Domain4 (CV2). The simulation reached

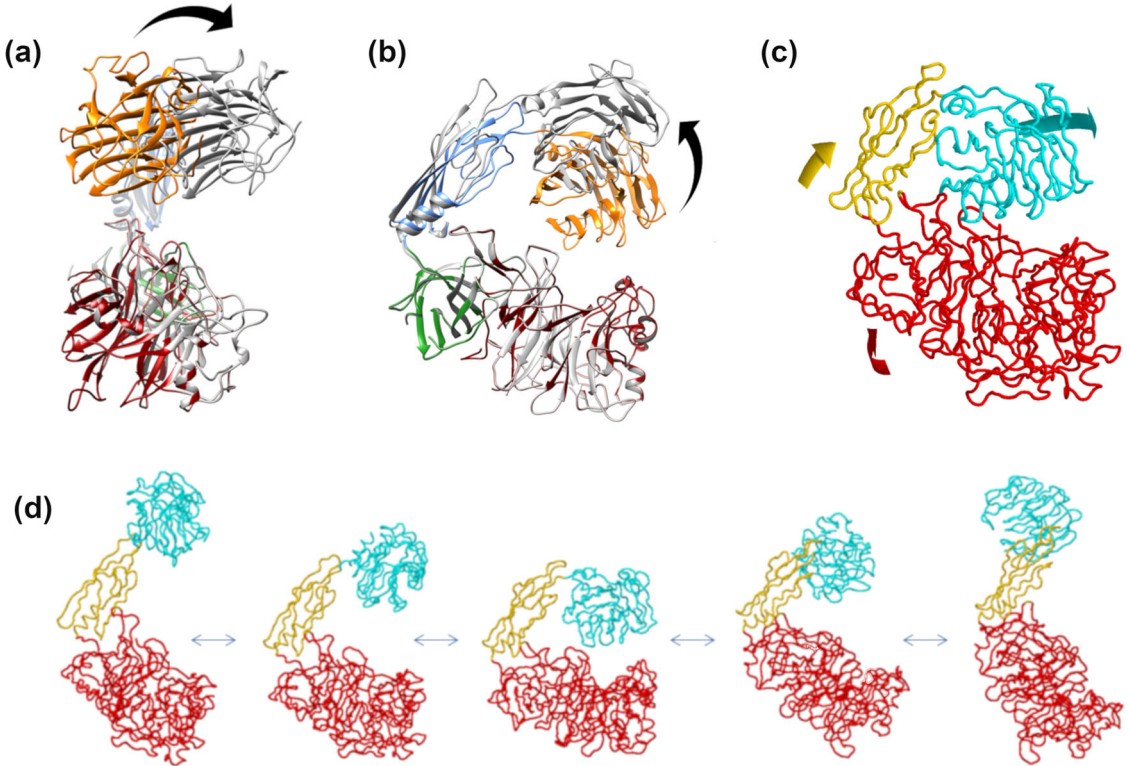

**Fig. 3 Prediction of conformational change by normal mode analysis (NMA).** The NomadRef server[44] predicted **a** a "sideways" motion of Domain4 (orange to gray) for the lowest frequency mode, and **b** an "upward" motion of Domain4 for the second lowest frequency mode. **c**, **d** The iModS webserver[45] predicted for the lowest frequency mode a combination of "sideways" and "opening-up" motions for Domain4.

convergence after 1.12 μs of simulation time (Supplementary Fig. 3), and its analysis with the toolkit METAGUI[46,47] produced a continuous free energy surface as a function of the two chosen CVs. The free energy surface revealed a considerably larger conformational space compared to the one sampled by the classical MD simulations, and specifically, three main local energy minima in the conformational space (EM1-EM3), as well as a small energy minimum shoulder located on an energetic plateau (EM4) (Fig. 4b). The two lowest energy minima correspond to relatively closed conformations, namely EM1 ($\Delta\Delta G = 0$ kcal mol$^{-1}$), which corresponds to a distance of 55 Å between Domain1 and Domain4 and a dihedral angle of $-63°$ between the four domains, and EM2, which is 1.2 kcal mol$^{-1}$ higher in energy than EM1 ($\Delta\Delta G = 1.2$ kcal mol$^{-1}$), and corresponds to respective CV values of 50 Å and 27°. EM3 ($\Delta\Delta G = 2.5$ kcal mol$^{-1}$) and EM4 ($\Delta\Delta G = 8.2$ kcal mol$^{-1}$) correspond to wider open conformations, namely to conformations with CV values of (67 Å, $-63°$) and (74 Å, 45°), respectively. Interestingly, the CV values of the crystal structure AbnA-Conf1 (48 Å, 8.0°) correspond to the EM2 minimum of the metadynamics energy landscape (Fig. 4b). The other crystallographic conformational states, AbnA-Conf2 (41 Å, 8.7°) and AbnA-Conf3 (42 Å, 7.0°), are located right at the border of the conformational space that was sampled by the metadynamics simulation, owing to the necessity of introducing upper and lower restraining potential walls into the CV1 collective variable to assist the simulation in reaching convergence (see Methods). Nevertheless, since all three crystallographic structures adopt a relatively "closed" structure, it can generally be assumed that the crystallographic structures correspond to the EM2 state.

**SAXS data support the metadynamics local minima structures.** To investigate further the conformational flexibility observed for AbnA through MD and metadynamics simulations, small angle

X-ray scattering (SAXS) data were collected for AbnA-WT and the truncation mutants AbnA-D123 and AbnA-D12 (Supplementary Fig. 4a). The radius of gyration ($R_g$) for AbnA-WT was calculated to be $36 \pm 1$ Å. Its pair distance distribution function P(r) demonstrated a wide distribution of most probable distances, with a main peak at a pairwise distance of 42.1 Å and an additional shoulder peak at ~55 Å (Supplementary Fig. 4b). In contrast, the $R_g$ values for the AbnA-D123 and AbnA-D12 truncation mutants were expectedly smaller than the WT ($32 \pm 1$ and $27 \pm 1$ Å, respectively), and their P(r) distributions were narrower, displaying a smaller distribution of most probable distances (Supplementary Fig. 4b). When comparing their conformational flexibility using the Porod-Debye plot[48], it can be seen that AbnA-WT displays considerable flexibility (as apparent from the complete lack of plateau in the plot[48]), whereas AbnA-D123 displays flexibility to a significantly lesser degree, and AbnA-D12 does not display such flexibility at all (as apparent from the clear plateau) (Supplementary Fig. 4c). These results are in line with the large conformational flexibility revolving around Domain3 and Domain4 that was observed for AbnA-WT by simulations; and as expected, when truncating Domain3 and Domain4, this flexibility is abolished.

The SAXS data were then used to generate averaged low-resolution envelope models for AbnA-WT, AbnA-D123, and AbnA-D12 with the program Dammin[49]. As apparent from Fig. 5a, Domain1 and Domain2 of the AbnA structure fits the envelope qualitatively rather well. On the other hand, when fitting the crystal structures of AbnA-D123 and AbnA-WT to the SAXS envelope, the fit is much worse (Supplementary Fig. 4d). However, when considering the flexibility of Domain3 and Domain4 by using representative structures from the metadynamics local minima (specifically from the EM3 and EM4 minima), much better fits are obtained for AbnA-D123 and AbnA-WT, though to a lesser extent

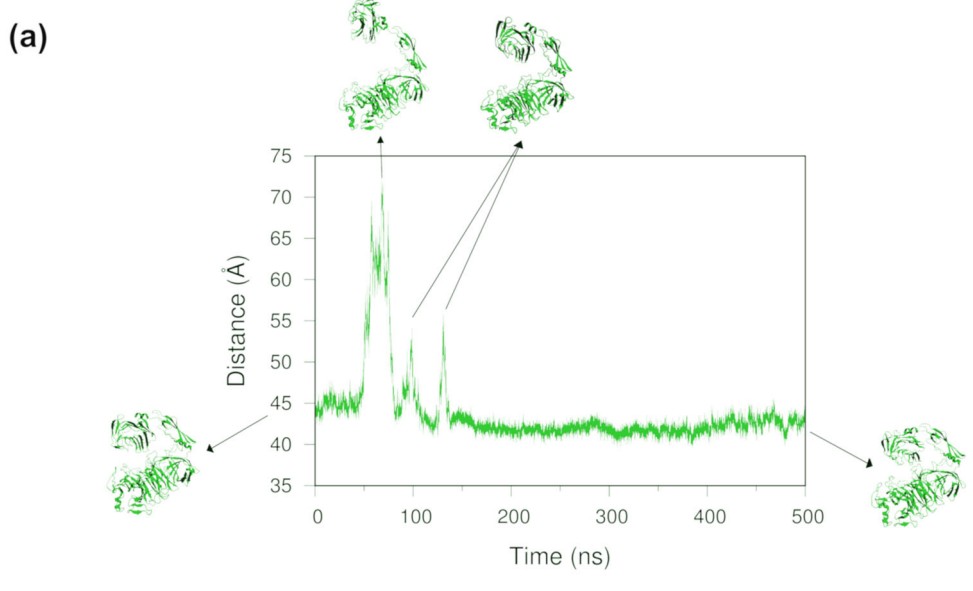

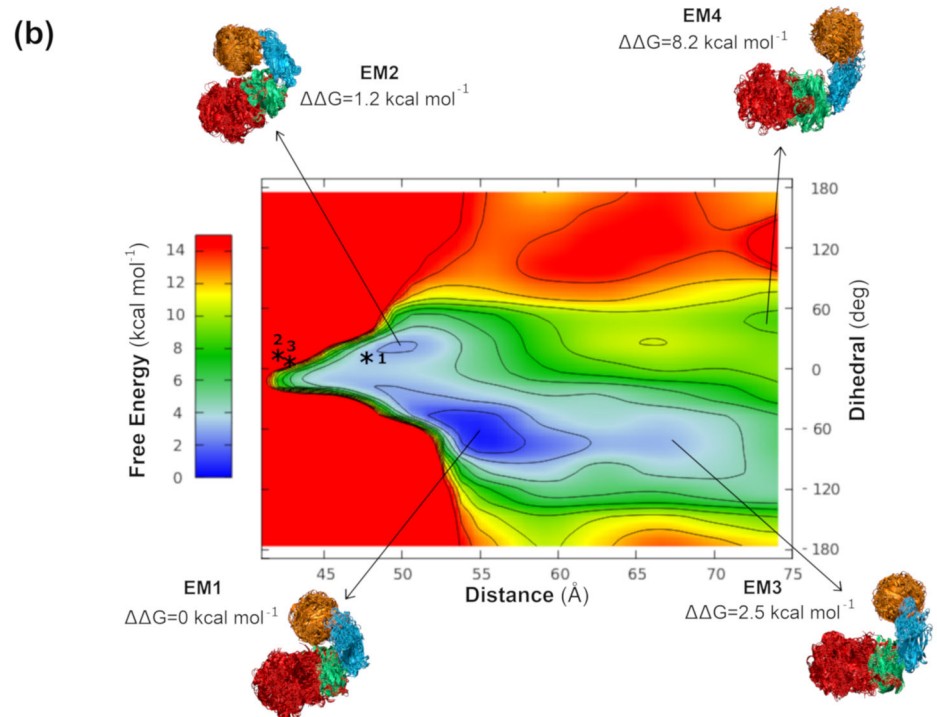

**Fig. 4 Large conformational changes detected by MD and metadynamics simulations. a** An all-atom classical MD simulation of the AbnA-Conf2 crystal structure over 500 ns. The distance between Domain1 and Domain4 is plotted as a function of time. **b** The free energy landscape obtained by metadynamics for the conformational dynamics of AbnA as a function of (i) the distance between Domain1 and Domain4 and (ii) the dihedral angle between Domain1, Domain2, Domain3, and Domain4. Coloring corresponds to differences in energy values ranging from 0 (blue) to 15 kcal mol$^{-1}$ (red). Contour lines are contoured at 1.4 kcal mol$^{-1}$. Three energy minima are observed, as well as a small energy minimum shoulder located on an energetic plateau. The structural clusters, representative of each energy minimum, are presented, and the relative difference in energy between them is shown. Stars 1,2,3 indicate the locations of the AbnA-Conf1, AbnA-Conf2, and AbnA-Conf3 crystal structures on the energy landscape.

for the WT compared to AbnA-D123 (Fig. 5a). The less good fit obtained when fitting the AbnA-WT SAXS envelope supports our previous metadynamics finding that the AbnA protein possesses multiple local minima conformations (Fig. 4b), so that the SAXS envelope represents an average of multiple conformations in solution, not just one conformation.

To obtain a more specific insight from such multiple-conformational fit to SAXS data, the program MultiFoXS[50] was

used, which takes into account conformational heterogeneity in solution through multi-state modeling with SAXS profiles. Such an approach is supported by the broad distribution of probable pairwise distances apparent from the SAXS P(r) function, and by the conformational flexibility that was apparent from the Porod-Debye plot (Supplementary Fig. 4b, c), which indicate multiple conformations in solution. The clusters pertaining to the four energy minima observed in the metadynamics free energy

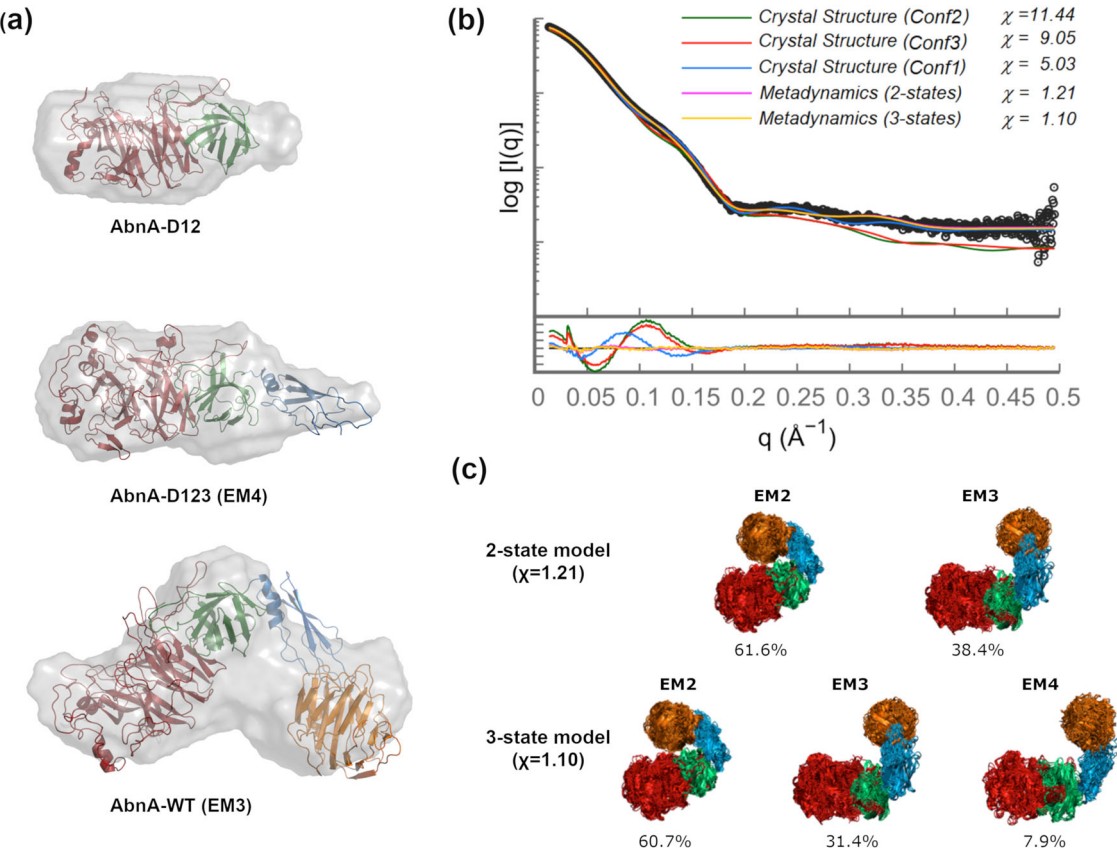

**Fig. 5 SAXS data agree with metadynamics results. a** Qualitative fit of the AbnA-D12 (taken from the AbnA-Conf1 crystal structure), AbnA-D123 (taken from the EM4 metadynamics cluster), and AbnA-WT (taken from the EM3 metadynamics cluster) structures inside their averaged SAXS envelopes. **b** Successful fitting of the AbnA-WT SAXS scattering curve (in black) is achieved when considering 2- or 3-state models composed of the clusters representative to the energy-minima obtained in the metadynamics free energy conformational landscape of AbnA, with population weights as shown in (**c**). In contrast, such a fit is much worse (high χ values) when considering only the crystallographic structures.

landscape were used to fit the AbnA-WT SAXS data. Using representative structures from each of the four clusters, a very good fit of $\chi = 1.21$ was obtained when assuming a 2-state model composed of the EM2 and EM3 representative structures (population weights of 61.6% and 38.4%, respectively). An even better fit of $\chi = 1.10$ was obtained when assuming a 3-state model composed of EM2, EM3, and EM4 representative structures (population weights of 60.7%, 31.4%, and 7.9%, respectively) (Fig. 5b, c). This is in contrast to χ values of 5.03–11.44 that were obtained when using only the crystal structures to fit the SAXS data (Fig. 5b). Interestingly, the population weights that were assigned to each representative metadynamics structure for the SAXS fitting correspond qualitatively to the energy differences between these clusters, as derived from the metadynamics free energy landscape, providing further support for the correlation between the metadynamics and the SAXS data (Figs. 4b, 5c). The structure representative of EM1, which is lowest in energy, was not present in the best multi-state fit. However, when using its structure to produce a 2-state model, composed 73.3% of EM1 and 26.7% of EM3, the χ value ($\chi = 2.31$) still decreases considerably. This suggests that since the EM1 and EM2 clusters are structurally similar in terms of their state of closure (both relatively "closed"), in the resolution required to fit the SAXS data, EM1 and EM2 are generally equivalent.

The metadynamics multi-state models obtained by the SAXS fitting correlate well with the SAXS $R_g$ value of 36 ± 1 Å. The $R_g$ calculated directly from the AbnA crystal structures is between 29.5–31.4 Å, and the $R_g$ values calculated from the metadynamics structures comprising the 3-state model are EM2 = ~32 Å,

EM3 = ~37 Å, and EM4 = ~38 Å. When considering the population weights obtained for the multi-state models, a calculated weighted mean of $R_g$ = ~34 Å is obtained. When this calculated value is taken into account with a hydration layer of 2–3 Å[51], a good correlation is obtained with the experimental value of 36 ± 1 Å derived from SAXS.

**Trapping the closed state validates the functional relevance of large conformational changes.** To investigate and validate further the biological relevance of the conformational changes of AbnA, we attempted to trap the enzyme in one of its "closed" states. To this end, we specifically changed residues Glu420 (Domain1) and Trp758 (Domain4) of AbnA-WT into cystein residues (resulting in **AbnA-E420C-W758C**), since these residues form a strong hydrogen bond between Domain1 and Domain4 in the AbnA-Conf2 conformation (Fig. 2b). Oxidized glutathione (GSSG) was added to AbnA-E420C-W758C as an oxidant, to assist in formation of a disulfide cross-link bridge between residues E420C and W758C.

Dynamic light scattering (DLS) was used to measure the hydrodynamic diameter of both AbnA-WT and AbnA-E420C-W758C in solution. The diameter of AbnA-WT was shown to be 16.9 ± 0.2 nm, while the diameter of the oxidized AbnA-E420C-W758C was 9.1 ± 0.3 nm. Interestingly, upon addition of the reducing agent DTT to the oxidized AbnA-E420C-W758C solution, the diameter extended from 9.1 ± 0.3 to 14.4 ± 0.2 nm, a significantly longer diameter that better resembles the one observed for the WT protein (Fig. 6a). A hydrodynamic diameter

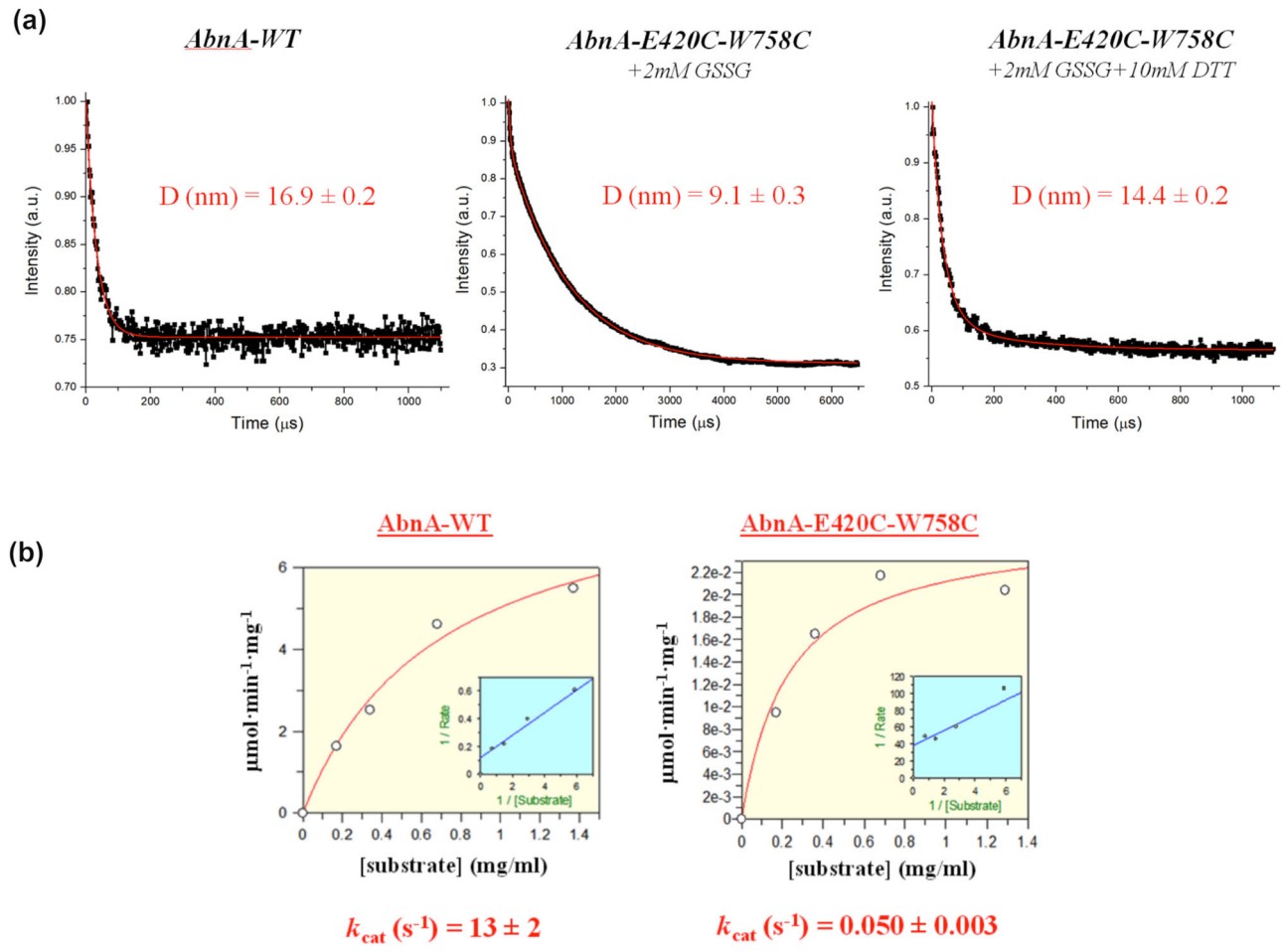

**Fig. 6 Two point mutations change the DLS diameter and kinetic activity of AbnA. a** DLS measurements reveal a hydrodynamic diameter of $16.9 \pm 0.2$ nm for AbnA-WT and a diameter of $9.1 \pm 0.3$ nm for AbnA-E420C-W758C, when oxidized by 2 mM GSSG. After the addition of 10 mM DTT to the oxidized AbnA-E420C-W758C, the diameter changed to $14.4 \pm 0.2$ nm. **b** Kinetic measurements of AbnA-WT ($k_{cat} = 13 \pm 2 \, s^{-1}$) and oxidized AbnA-E420C-W758C ($k_{cat} = 0.050 \pm 0.003 \, s^{-1}$) on branched arabinan demonstrate a x260 reduction in the catalytic activity.

of $9.1 \pm 0.3$ nm corresponds closely to the dimensions of the "closed" crystal structure, which are $\sim 80 \times 70 \times 30$ Å. The presence of a $16.9 \pm 0.2$ nm hydrodynamic diameter, as measured for AbnA-WT, suggests a more extended conformation for AbnA in solution, possibly resembling the wider extended conformations observed in the metadynamics simulations, as were also confirmed by the SAXS results. The fact that addition of DTT served as a switch to convert the hydrodynamic diameter of AbnA-E420C-W758C from $9.1 \pm 0.3$ to $14.4 \pm 0.2$ nm indicates that a disulfide cross-link bridge, formed by the double mutation, indeed locked AbnA-E420C-W758C through an S-S cross-linking into a closed conformation, and that the AbnA-Conf2 conformation indeed exists in solution.

The activities of AbnA and AbnA-E420C-W758C towards branched arabinan were essentially the same when the proteins were used in a standard buffer. Thus, the two Cys replacements did not affect activity. However, the addition of 5 mM GSSG markedly affected the activity of AbnA-E420C-W758C. The $k_{cat}$ for the WT enzyme was $13 \, s^{-1}$, while for the oxidized E420C-W758C mutant it was $0.05 \, s^{-1}$ (Fig. 6b). Thus, the activity of the oxidized AbnA-E420C-W758C was lowered by about two orders of magnitudes compared with both the WT and the non-oxidized E240C-W758C mutant. This confirms that the observed reduction in activity is not just the result of the double mutation alone, and suggests that the significant activity drop in the oxidized (and cross-linked) mutant resulted mainly because the substrate could

not access the active site while Domain4 was covalently bound to the catalytic Domain1.

## Discussion

The structure of AbnA is remarkably larger than any other structure solved so far from the GH43 family. Crystal structures of full-length ultra-multimodular GHs have been solved in the past, primarily from the GH13 family[26,28,29,33], but the structure of AbnA is the only full-length ultra-multimodular structure to be solved within the GH43 family. The structure of AbnA adopts a "pincer-like" structure built of four distinct domains instead of the one or two domains usually found in enzymes from this family. Domain1 of AbnA corresponds to the five-bladed propeller structure typical of GH43 enzymes, and is thus assigned to be the catalytic domain of the enzyme. Indeed, by using soaking experiments, an arabinooctaose substrate was successfully trapped in the presumed active site of Domain1, using a truncation mutant lacking Domain4 (AbnA-D123-A8). The arabinooctaose molecule binds a long groove that spans the full length of Domain1, enabling complete mapping of the eight binding subsites (Fig. 1c, Supplementary Fig. 1b).

Domains that resemble Domain2 have been observed in the structures of only two other GH43 arabinanases[36,37]. Domain2 forms many interactions with the catalytic Domain1, and seems to act as its extension. In fact, although not fully apparent from

homology sequence alignment, Domain2 appears to bear a general structural resemblance to the CBM2 family fold (Supplementary Fig. 5a), and may thus provide an "extended platform" for binding longer substrates at the active site. Although the A8 substrate that was trapped in the active site of AbnA-D123 does not seem to interact directly with Domain2 and according to this structure interacts mainly with Domain1 (Supplementary Fig. 5b), the considerably larger and more complex native arabinan substrate may need a larger platform (as extended by Domain2) to which it can better bind. Indeed, it has been demonstrated in the past how a CBM domain, positioned in proximity to the catalytic domain, may form part of the active site and bind substrate[52]. Thus, Domain2 in AbnA (and in other homologs from the GH43 family[36,37]) may adopt a similar role, thereby representing a new uncharacterized CBM family.

The structures of domains analogous to Domain3 and Domain4 have not been reported before for other GH43 arabinanases. The fold of Domain4 also bears resemblance to a CBM domain (as detailed in the Results section), and taken together with ITC experiments, suggests that this domain may also belong to a new uncharacterized CBM family. CBMs that recognize one or two monosaccharide units of arabinose have been reported in the past in the CBM13 and CBM42 families[53–56], but as far as we know, Domain4 (and possibly also Domain2) represent the first reported cases of CBM domains that bind longer arabino-oligomers and arabinan polymer[57].

Two additional conformational states were determined for AbnA by X-ray crystallography, and this conformational flexibility was then further investigated thoroughly through the combined use of MD, metadynamics, NMA, SAXS, DLS, cross-linking, and kinetic experiments. Although this work is the first report where such large conformational changes have been demonstrated for an enzyme from the GH43 family, such functional global flexibility seems to be typical to many other ultra-multimodular enzymes. Thus, for example, it has been determined through the combined use of X-ray crystallography and ensemble optimization modeling (EOM) to SAXS data[25], how the glycoside hydrolases MmChi60[31] from *Moritella marina* (family GH18), Sp3GH98 and Sp4GH98[30] from *Streptococcus pneumonia* (family GH98), SpuA[33] from *Streptococcus pneumonia* (family GH13), and TpMan[32] from *Thermotoga petrophila* (family GH5) appear to also undergo similar large conformational changes. In this work, however, we demonstrate a more thorough and detailed approach to investigate such large conformational changes, through the combined use of MD, metadynamics, NMA, cross-linking, and DLS experiments, in addition to the use of X-ray crystallography and SAXS data. Such an integrative approach is also known as the "integrative structure determination" approach, whereby one combines multiple complementary methods, both experimental and computational, to obtain a comprehensive understanding of protein structure and dynamics at the atomic level[58–60]. Such an approach allows more realistic and energetically-based information to be obtained regarding the conformations observed in solution, since the sampling of the wide-open conformations are scored energetically by other methods (in this case by MD, metadynamics, and NMA calculations), which provide more realistic information regarding the energetic stability of the sampled conformations. Furthermore, in the present work, besides collection of SAXS data, the addition of DLS, cross-linking, and kinetic experiments provide experimental

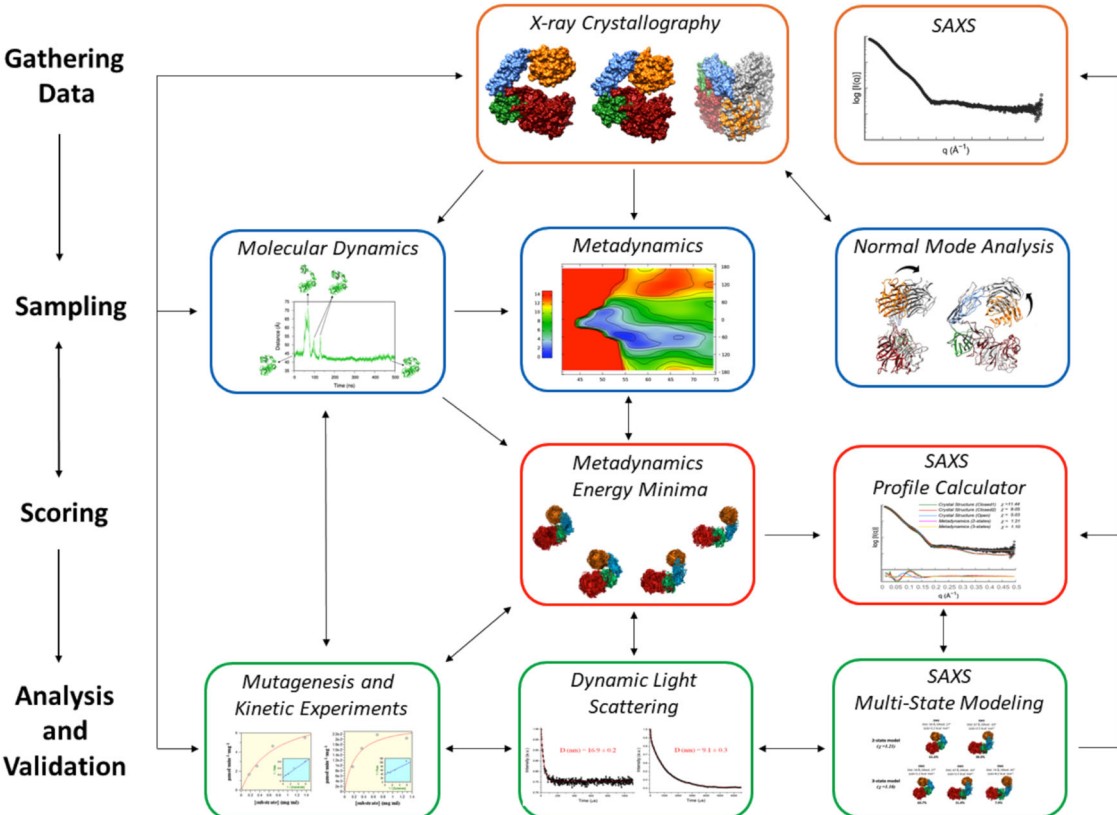

**Fig. 7 The integrative structure determination of AbnA.** The integrative structure determination of AbnA was conducted through four stages: (1) gathering of X-ray crystallography and SAXS data, where three crystallographic conformational states were determined; (2) sampling additional conformational states using normal mode analysis, molecular dynamics, and metadynamics; (3) scoring the energetic and biological relevance of these conformational states with metadynamics and SAXS data; (4) validating the large conformational changes sampled and scored in stages (2) and (3) with SAXS, dynamic light scattering (DLS), mutagenesis, and kinetic experiments.

validation to the biological relevance and functionality of these conformations (Fig. 7).

The large conformational changes observed for AbnA are in line with the function of Domain4 as a CBM domain. The role of a CBM is mainly to assist the catalytic domain in reaching the relatively scarce and insoluble substrate outside the microbial cell by detecting and binding to it, and by doing so, increase the effective concentration of the enzyme near it[38]. This function of AbnA is facilitated by the ability of the enzyme to undergo the large functional conformational changes demonstrated here, allowing AbnA to sample more space and increase its probability to find and bind the insoluble polysaccharide substrate in the environment. Such a role for Domain4 is interestingly supported by the fact that the orientation of the substrate bound in the active site of AbnA is perpendicular to the orientation of the substrate bound to Domain4 (Fig. 1d). This implies that the oligosaccharide strand onto which Domain4 binds is not necessarily the same strand that the catalytic Domain1 hydrolyses. Alternatively, however, it can be hypothesized that AbnA undergoes such large conformational changes in order to hydrolyze its substrate through a "harpoon" mechanism. By this mechanism, AbnA may equilibrate within the wide spectrum of conformations it possesses until the arabinan is sensed by Domain4. Once sensed, Domain4 binds the arabinan like a harpoon, bringing the catalytic domain after it (through conformational change) for the rapid and efficient degradation of the arabinan. Obviously, the validity of such a mechanism awaits further study and experimental evidence.

To conclude, we determined in this work the first full-length experimental structure of an ultra-multimodular enzyme from the GH43 family (AbnA), possessing two domains that may represent two new uncharacterized CBM families. We investigated thoroughly the conformational flexibility of AbnA using the "integrative structure determination" approach, demonstrating that this protein undergoes large functional conformational changes. Approaches similar to the integrated approach used here may be used in the future to investigate in great detail other ultra-multimodular proteins of diverse structures and functions.

## Methods

### Cloning, overexpression, and purification of AbnA and its mutants.
AbnA and its mutant derivatives were overexpressed efficiently in *E. coli* BL21(DE3) carrying the pET9d plasmid with the desired gene, using the T7 polymerase expression system. All AbnA variants were cloned without the leader peptide (28 residues), and with the addition of an N-terminal 6xHis tag. The *E. coli* culture was grown on TB with kanamycin, the leaky promoter of the T7 polymerase allowed sufficient expression of the target proteins without induction, when the cells were grown on TB medium. Cells from 0.5 lit culture were put on ice for 15 min, harvested by centrifugation (7000 rpm for 5 min), and resuspended in binding buffer (20 mM imidazole, 0.5 M NaCl, and 20 mM phosphate buffer, pH 7.4). The cells were disrupted by two passages through an Avestin Emulsiflex C3 Homogenizer, 1500 psi (Avestin) at room temperature and centrifuged (14,000 rpm for 30 min at 15 °C). The soluble fraction, containing the protein, was subjected to heat treatment at ~45 °C for half an hour and centrifuged (14,000 rpm for 30 min at 15 °C). The His-tagged fused protein was purified using the ÄKTA Avant-25 chromatography system (GE Healthcare Life Sciences) equipped with a HisTrap column (1 or 5 ml column volume, GE Healthcare Life Sciences), according to the manufacturer's instructions. The proteins were eluted with a 10 CV gradient of elution buffer that contained 0.5 M imidazole, 0.5 M NaCl, and 20 mM phosphate buffer, pH 7.4. Typically, between 50 and 200 mg of the recombinant proteins were obtained from one liter of over-night *E. coli* culture and based on SDS–PAGE, the proteins were about 95% pure. The purified protein was then dialyzed against a solution of 50 mM Tris-HCl buffer, pH 7.0, 100 mM NaCl, and 0.02% sodium azide, used for the subsequent crystallization experiments.

### AbnA-WT crystallization and data collection.
The purified AbnA-WT protein (11 mg/ml), in a buffer solution of 50 mM Tris-HCl pH 7.0 and 100 mM NaCl, was screened for crystallization conditions using an extensive series of factorial screening solutions set up in 96-3 iQ TTP Labtech plates with the *mosquito* LCP crystallization robot (TTP Labtech). Initial crystals were obtained from condition #31 of the PACT screen of Qiagen (Qiagen, Hilden, Germany). These were reproduced and refined in

4×6 VDX crystallization plates (Hampton Research, CA, USA) using the micro-seeding technique[61], resulting in rice-shaped crystals grown in drops consisting of 3 μl protein solution, 2 μl reservoir solution (17–19% PEG 6 K, 0.2 M NaCl, 0.1 M HEPES buffer, pH 7) and 1 μl micro-seeding solution. These AbnA-WT crystals were then used for a number of X-ray diffraction data measurement sessions with synchrotron radiation (ESRF, Grenoble, France). Using two different cryo-protecting solutions (10% PEG400 (**AbnA-PEG400**) or 10% ethylene glycol (**AbnA-EtGly**) in 90% reservoir solution), two full diffraction data sets were obtained ($\lambda = 0.98$ Å; 100 K), both belonging to the orthorhombic space group $P2_12_12_1$ but with slightly different unit cell dimensions. Please note that the AbnA-PEG400 structure is later termed **AbnA-Conf1**, and similarly the AbnA-EtGly structure is later termed **AbnA-Conf2** (Table 1). Processing, reduction, indexing, integration and scaling of the diffraction data were conducted using the HKL-2000 suite[62], and data collection details and statistics are summarized in Table 1.

### Structure determination of AbnA-Conf1.
The AbnA-PEG400 dataset (which will later be termed AbnA-Conf1) was used first for molecular replacement (MR) calculations, using as search model the structure of the endo-1,5-α-L-arabinanase from *Bacillus Subtilis* (PDB code 2X8T;[37] 470 AA; 42% identity over 63% of the AbnA sequence). The program Phaser[63] gave an MR solution of one AbnA monomer in the crystallographic asymmetric unit (AU) (42.4% solvent content, as estimated by Matthews Coefficient calculations[64]), with a log-likelihood gain (LLG) of 582, a translation function Z-score equivalent (TFZ) of 8.4, and initial $R_{factor}$ and $R_{free}$ of 42.9% and 50.2%, respectively. The coordinates and maps obtained from this solution were then used as input files for the AutoBuild Wizard program[65]. The model and maps obtained from this first Autobuild cycle showed improved $R_{factor}$ and $R_{free}$ values of 38.5% and 45.1%, respectively. However, the model was still missing ~350 AA, which were then manually built and fitted into electron density map with the program Coot[66], followed by concomitant refinement with Refmac5[67]. Every few cycles of manual building and refinement were followed by an automatic building cycle in Autobuild[65].

After eight such Autobuild cycles, the model contained almost all residues, except for 24 residues at the N-terminus and 3 residues at the C-terminus, with final $R_{factor}$ and $R_{free}$ values of 16.5% and 24.2%, respectively (Table 1). Validation of all structural and stereochemistry parameters was done with PROCHECK[68], showing that 96.5% residues were in the "most favored" regions of the Ramachandran plot, 3.4% in the "additionally allowed" regions, and 0.1% residues in the "disallowed" regions. Further structural and refinement parameters are summarized in Table 1.

### Structure determination of AbnA-Conf2.
The AbnA-EtGly dataset was solved using the already solved AbnA-PEG400 structure. The structure determination process was not straightforward, and was only possible after dividing the AbnA-PEG400 search model into two independent models, one model composed of Domain1, Domain2 and Domain3 of the AbnA structure, and the other composed of Domain4. Phaser[63] gave a solution with initial $R_{factor}$ and $R_{free}$ values of 23.3% and 29.8%, respectively, and subsequent rebuilding and refinement with Coot[66] and Refmac5[67] reduced these values to 17.8% and 25.5%, respectively. Analysis with PROCHECK[68] revealed that 95.5% residues were in the "most favored" regions of the Ramachandran plot, 4.4% in the "additionally allowed" regions, and 0.1% residues in the "disallowed" regions. Additional structural and refinement parameters for this structure are summarized in Table 1. Comparison of the resulting AbnA-EtGly structure with the original AbnA-PEG400 structure reveals two different conformational states, as discussed in the text. The structure of AbnA-EtGly is therefore termed **AbnA-Conf2**, and the structure of AbnA-PEG400 is termed **AbnA-Conf1**.

### Structure determination of AbnA-Conf3.
A second closed conformational state was observed for AbnA following soaking experiments with arabinoheptaose (A7). In these experiments, AbnA-WT crystals were soaked for 7–30 s in a cryo solution containing 10% PEG400 and 2 mM A7 in 90% well solution prior to flash-cooling in liquid nitrogen. Data to 2.19–2.89 Å were then collected on these crystals ($\lambda = 1.00$ Å; 100 K) at the DESY synchrotron (P11 beamline, PETRAIII, Hamburg, Germany), and XDSapp[69] was used for data processing. Data collection details and statistics are summarized in Table 1. MR calculations with Phaser[63], using Domain1, Domain2, Domain3 and Domain4 of AbnA-Conf2 as separate search models, produced a closed conformation of a different orientation. Refinement with Coot[66] did not reveal any extra density in the structures corresponding to an arabino-oligosaccharide. A representative structure corresponding to a resolution of 2.19 Å was refined with phenix.refine[70] to $R_{factor}$ and $R_{free}$ values of 19.2% and 25.2%, respectively. Analysis with PROCHECK[68] revealed that 96.2% residues were in the "most favored" regions of the Ramachandran plot, 3.4% in the "additionally allowed" regions, and 0.4% residues in the "disallowed" regions. Additional structure refinement and validation parameters are summarized in Table 1.

### Structure determination of AbnA-WT-A5.
The structure of AbnA in complex with arabinopentaose (**AbnA-WT-A5**) was obtained by soaking an AbnA-WT crystal in a cryo solution (10% PEG400 in 90% well solution) containing 5 mM arabinopentaose for ~30 s prior to flash-cooling for data collection. Data were collected at the ESRF synchrotron ($\lambda = 0.98$ Å; 100 K), and processing with the

HKL-2000 suite[62] produced a complete dataset to 2.95 Å resolution. Data collection details and statistics are summarized in Table 1. Structure determination was achieved by molecular replacement with Phaser[63], using the structure of AbnA-Conf1 as search model (the structure of AbnA-Conf2 and AbnA-Conf3 did not give a clear solution). Refinement of the data revealed extra density in Domain4, which was modeled in Coot[66] as an arabinopentose molecule of 70% occupancy. Extra density was also observed in the active site, but this did not correspond to an oligosaccharide, and was modeled instead as a Tris and a diethylene glycol (DEG) molecule. After a few cycles of refinement the $R_{factor}$ and $R_{free}$ values converged to 16.8% and 25.9%, respectively. Analysis with PROCHECK[68] revealed that 93.3% residues were in the "most favored" regions of the Ramachandran plot, 6.4% in the "additionally allowed" regions, and 0.4% residues in the "disallowed" regions. Additional structural and refinement parameters are summarized in Table 1.

**Structure determination of AbnA-D123-A8**. Crystals of AbnA-D123 (truncated AbnA, Domain1, Domain2, and Domain3) co-crystallized with arabinooctaose (**AbnA-D123-A8**) were obtained using the hanging drop method, grown in 5 µl drops containing 3 µl protein solution (8.3 mg/ml in 50 mM Tris buffer pH 7, 100 mM NaCl, 0.02% NaAzide), 1.5 µl reservoir solution (1.8–2.0 M AmSO$_4$, 1% PEG 400, 0.1 M Hepes pH 7.5), and 0.5 µl arabinooctaose solution (50 mM). Diamond-shaped crystals of roughly $0.5 \times 0.2 \times 0.2$ mm dimensions were obtained after 3–7 days. A full dataset was collected on one of these crystals at the DESY synchrotron ($\lambda = 0.98$ Å; 100 K) following a quick soak in a cryo solution containing 20% PEG400 (80% well solution) before flash-freezing in liquid nitrogen. Processing with XDSapp[69] produced a dataset to 2.84 Å resolution belonging to the P6$_1$22 space group, of unit cell dimensions $a = b = 129.2$ Å, $c = 488.4$ Å. Data collection details and processing statistics are summarized in Table 1. MR with Phaser[63], using Domain1, Domain2 and Domain3 of AbnA-Conf2 as search model, gave a solution of two monomers in the asymmetric unit. Extra density corresponding to arabinooctaose molecules was observed in the two active sites, and was therefore modeled as such in Coot[66] and refined with phenix.refine[70] to final occupancies of 71% and 75%. Refinement converged to $R_{factor}$ and $R_{free}$ values of 18.3% and 24.2%, respectively. Analysis with PROCHECK[68] revealed that 96.0% residues were in the "most favored" regions of the Ramachandran plot, 4.0% in the "additionally allowed" regions, and 0.1% residues in the "disallowed" regions. Additional structural and refinement parameters are summarized in Table 1.

**ITC measurements**. Isothermal titration calorimetry measurements were performed at 30 °C with a MicroCal iTC200 titration calorimeter (Malvern Instruments Ltd, UK). Protein solutions for ITC were dialyzed extensively overnight against buffer (50 mM Tris-HCl, 100 mM NaCl and 0.02% NaN$_3$, pH 7.0). Ligand solutions of sugar-beet arabinan and linear arabinan were prepared by dissolving them with the actual protein dialysis buffer. Aliquots (1.5–2 µl) of the ligand solution, at least 10 times higher than the molar concentration of the binding-sites, were added to the reaction cell containing 200 µl of (0.14–0.35) mM protein solution by the controlled action of a 40 µl rotating stirrer-syringe. The estimated molar concentration of binding sites on the polysaccharides was derived by manually altering the value to reach $n = 1$ ($n$-number of binding sites). The heat of dilution was determined to be negligible in separate titrations of the ligand into the buffer solution. Calorimetric data analysis was carried out with the Origin 7.0 software (OriginLab). Binding parameters, including the binding constant ($K_B$, M$^{-1}$), and the binding enthalpy ($\Delta H_B$, kcal/mol of bound ligand), were determined by fitting of the experimental binding isotherms. Results for these measurements are presented in Table 1 and Supplementary Fig. 2.

**Molecular dynamics and metadynamics simulations**. The all-atom classical MD simulation was run using Gromacs2016[71]. The simulation was performed using the Amber03 force field[72] with the Tip3P water model[73], and the starting conformation was taken from the AbnA-Conf2 crystal structure. This structure was solvated with ~50,000 water molecules and neutralized with Na$^+$ and Cl$^-$ ions to a concentration of 150 mM in a dodecahedron box, with a minimal distance of 2.0 nm between protein and box wall. The van der Waals interactions were implemented with a cutoff at 1.0 nm, and long-range electrostatic effects were treated with the particle mesh Ewald method. The protein-solvent model was then put through 10 rounds of geometry optimization and energy minimization, followed by a 100 ps protein position-restrained equilibration and an additional 100 ps of unrestrained equilibration. The system was then heated to 300 K at incremental steps of 25 K every 5 ps using the Noose-Hoover thermostat, and then equilibrated to a constant pressure of 1 bar using the Parrinello-Rahman barostat. Following these equilibration procedures, a time trajectory of 500 ns was simulated (Fig. 4a) at constant temperature and pressure, using time steps of 2 fs and the same thermostat and barostat. Analysis of the trajectory was performed with Gromacs, VMD, and Plumed Driver[74,75].

The metadynamics simulation was performed using the same setup and equilibration procedures as for the all-atom classical MD simulation of AbnA-Conf2, but with simulation parameters modified with respect to two collective variables (CV) using Plumed 2.3[74]. These two collective variables included the distance between the centers of mass of Domain1 and Domain4 (CV1), and the dihedral angle between the centers of mass of Domain1, Domain2, Domain3, and Domain4 (CV2). Gaussians of a height of 0.289 kcal/mole and widths of 0.2 Å and

0.07 rad for the respective CVs were deposited every 20 ps as bias into the energy potential, and using this bias, the simulation was extended up to 1.12 µs. In order for the simulation to reach convergence, an upper and lower wall were defined after 120 ns of simulation for the CV1 collective variable as a restraining potential, at an upper distance of 75 Å and a lower distance of 41 Å, with a force constant of 0.36 kcal/Å$^2$ and a power of 2. Analysis of the metadynamics data was performed using METAGUI[46,47], a visual analysis toolkit implemented for VMD[75]. Using this program, the bidimensional CV space was discretized into 20 bins in each direction, enabling a division of the conformations explored during the simulation into 353 different microstates. The equilibration time was set to 350 ns at which the two collective variables start diffusing for the rest of the simulation (see Supplementary Fig. 3). The relative free energies ($\Delta\Delta G$) between the microstates were successfully determined for 290 contiguous microstates. The statistical convergence of this free energy estimations was assessed following the procedure described in ref. [76]. The energy estimation tolerance was initially set to 5 kcal/mol, producing a free energy surface projected onto the bi-dimensional clustering grid (Fig. 4b). The energy minima located in this bi-dimensional free energy surface could still be observed with the same proportional energy differences between them, and statistical convergence, at an energy tolerance of 1.8 kcal/mol.

**SAXS measurements**. Small angle X-ray scattering (SAXS) data were measured at the BM29 BioSAXS[77] beamline of ESRF for the three variants AbnA-WT, AbnA-D123, and AbnA-D12 (containing 50 mM of Tris/HCl pH 7.0 and 100 mM NaCl) (Supplementary Fig. 4a) at three different concentrations each, ranging from 1.3 to 9.0 mg/ml. For buffer subtraction, an identical buffer sample was prepared (without protein), and its corresponding SAXS profile was measured immediately before measurement of the protein sample. The measurements were performed with a Pilatus 1 M detector at 293 K, using a $0.7 \times 0.7$ mm X-ray beam of $\lambda = 0.992$ Å. Each experiment included 10 frames of 1 s exposure, performed on a 40 µl sample, which was flowing through a 1.8 mm diameter capillary. Initial data processing (background subtraction, radial averaging, etc.) was performed using the dedicated beamline software BsxCuBE (Biosaxs Customized Beamline Environment)[77]. The final scattering profile used for structure analysis (Fig. 5a) was obtained after merging the profiles that displayed no aggregation, according to the Guinier plot analysis (adj $R^2 > 0.995$ in the range $qr_g < 1.3$)[78]. The PRIMUS[79] software was used for merging of the data, while the GNOM[80] software was used for calculation of the pair distribution function P(r) (Supplementary Fig. 4b). Molecular envelopes for each AbnA variant were calculated by averaging and filtering 50 independent ab initio models constructed from the P(r) function, using the programs DAMMIN and DAMAVER[49,81] with default parameters, and using the program Supcomb[82] to fit the structures to the envelope. Fitting of the AbnA-WT structural data to the AbnA-WT SAXS profile was conducted with the program FoXS[50], and the multi-state fitting of the lowest energy metadynamics clusters to the SAXS data was conducted with the program MultiFoXS[50]. To produce an optimal fit of the various WT structures to the SAXS data, the program MODELLER[83] was used to model the N- and C- terminal fragments that were missing from the crystal structures, using the loop optimization algorithm with default parameters.

**DLS measurements**. The hydrodynamic diameter of AbnA-WT and AbnA-E420C-W758C was evaluated by a DLS instrument (Cordouan Technologies, VASCO™ -2 Particles Size Analyzer), equipped with a 65 mW diode laser, operating at 658 nm wavelength at a scattering angle of 135°. Samples were measured after a dilution to ~1 mg/ml at 25 °C. The results were analyzed with the nanoQ software, using the Pade-Laplace algorithm[84] for fitting of the obtained correlation function.

**Kinetics measurements**. The kinetic measurements for AbnA-WT, AbnA-E420C-W758C and AbnA-D1 on the native substrates sugar-beet arabinan and linear arabinan were performed using the BCA assay for reducing sugar determination[85]. The reactions were performed at 40 °C in 100 mM phosphate buffer: 19.5 ml 0.2 M NaH$_2$PO$_4 \cdot$H$_2$O and 30.5 ml 0.2 M Na$_2$HPO$_4 \cdot 2$H$_2$O in pH 7.0. The reactions were initialized by adding 75 µl of appropriately diluted, pre-warmed enzyme solutions, to 475 µl of pre-warmed substrate mixtures. During the initial stage of the reaction (12–15 min), aliquots (100 µl) were taken and placed in microtubes with 50 µl 1 M sodium carbonate solution. When the reaction was done, 150 µl of freshly made BCA reagent was added to each tube. The tubes were vortexed and placed in a water bath for 15 min incubation at 90 °C. After the samples cooled down, 200 µl from the samples were loaded to a 96-well plate and the absorbance was read at 560 nm using synergy HT plate reader. Values of $K_m$ and $k_{cat}$ were determined by non-linear regression analysis using the program GraFit 5.0. The ratio between arabinose concentration and the absorbance at OD 560 nm was determined following the same procedure above, with known arabinose concentrations that gave a linear calibration curve.

**Reporting summary**. Further information on research design is available in the Nature Research Reporting Summary linked to this article.

## Data availability

The AbnA sequence is available in Uniprot under accession number B3EYN2. The atomic coordinates of the AbnA-Conf1, AbnA-Conf2, AbnA-Conf3, AbnA-Conf1-A5,

and AbnA-D123-A8 are available in the Research Collaboratory for Structural Bioinformatics (RCSB) Protein Data Bank under accession codes 5HO2, 5HO0, 5HP6, 5HOF, and 5HO9, respectively. All other data are available from the corresponding authors upon reasonable request.

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

## Acknowledgements

This work was supported by the Israel Science Foundation Grants 500/10, 152/11 and 1905/15, the I-CORE Program of the Planning and Budgeting Committee, the Israeli Ministry of Environmental Protection, the Israeli Ministry of Science and Technology (MOST) grant No. 3-12484, and the Grand Technion Energy Program (GTEP). This work comprises part of The Leona M. and Harry B. Helmsley Charitable Trust reports on Alternative Energy series of the Technion, Israel Institute of Technology, and the Weizmann Institute of Science. The research also received funding from the European Community's Seventh Framework Program (FP7/2007-2013) under BioStruct-X (grant agreement N°283570). Y.S. acknowledges partial support by the Russell Berrie Nanotechnology Institute and The Lorry I. Lokey Interdisciplinary Center for Life Science and Engineering, Technion. D.S. acknowledges supported by the ISF grant 1466/18. S.L. is grateful to the Azrieli Foundation for the award of an Azrieli Fellowship. We thank the HPC-Europa 3 program and the Barcelona Supercomputing Center (BSC) for providing access to the BSC supercomputing resources. We thank the staff at the European Synchrotron Research Facility (ESRF, BM14 beamline) and DESY (P11 beamline) synchrotrons for their helpful support in the X-ray synchrotron data measurement and analysis. We also thank and acknowledge the helpful comments of the editor and the anonymous referees of this paper. The synchrotron experiments at ESRF were supported also by the ESRF internal funding program. Y.S. holds the Erwin and Rosl Pollak Academic Chair at the Technion.

## Author contributions

S.L., R.S., Y.S., and G.S. planned the experiments; S.L. and O.S. performed the crystallographic experiments. R.S. produced the proteins and performed the biochemical experiments; S.L. performed the structural and NMA analysis. S.L. and X.B. performed the M.D. and metadynamics experiments. S.L. and R.S. performed the DLS experiments. S.L. and D.S. performed the SAXS experiments and analysis. G.S., Y.S., and A.P. conceived and coordinated the research project; S.L. and G.S. wrote the paper.

## Competing interests

The authors declare no competing interests.
