## [Peer Review File · Communications Biology]

Reviewers' comments:

Reviewer #1 (Remarks to the Author):

This article describes primarily the structural analysis of a multimodular GH43 arabinanase, AbnA. The main outcomes of note are the discovery of a new carbohydrate-binding module and a full analysis of the conformational transitions of the enzyme. This is a technically very sound study with appropriate interpretations that I very much enjoyed reading and reviewing. There are in my opinion, however, some issues that I believe the authors should address.

In general, the tone of the article comes across as if the approach the authors have used to examine the conformational heterogeneity of a CAZyme is novel. This may have not been intentional, but the near complete lack of acknowledgement of similar studies performed in the past would make this seem the case to the uninitiated. I do believe the authors have used a particularly thorough suite of approaches; however, the examination of solution conformations of CAZymes is not new. X-ray crystallography (XRC) in combination with solution methods, largely focusing on SAXS but also including NMR, DLS, etc., has been applied to multimodular CAZyme for >15 years. In fact, a recent (2017) methods paper by Czjzek and Ficko-Blean outlines approaches to combine XRC with SAXS to examine multimodular CAZymes. A fairly large, but not complete, group of articles focusing on solution conformations of CAZymes are cited in that methods paper. These two authors have been quite influential in this area. As far as I can tell, the present study has failed to draw upon or acknowledge a fairly rich history of efforts in this field (more on this below).

In keeping with the above comment, the authors' apparent failure to incorporate the approaches and outcomes of similar past studies in their manuscript also means their results are not placed in the greater context of CAZyme research, and so it is not clear to the reader what the general significance may be. Indeed, the article by Lammerts van Bueren et. al. (*Structure*, 2011, 19:640–651) reveals the overall structure of a GH13 enzyme and its conformational transitions to be eerily similar to those reported by the authors. In comparison to this and other previous CAZyme studies that utilize SAXS, the relatively uncharacterized *M. viridifaciens* GH33 example that the authors use as a comparator seems very poor.

So, to summarize the previous two points, I believe the authors of this study have missed an opportunity to leverage a reasonably large body of existing literature in their study. To do so would not only informationally enrich their manuscript by providing more context but it would also help illuminate how the present approach may build upon past efforts. This would help elevate the interest because, at present, a skeptic could view the manuscript as it is presented as simply a very detailed study of a single esoteric enzyme. This reviewer would like to think there is more to it than this.

On the basis of the conformations adopted by AbnA the authors propose a "harpoon" mechanism. I find this imaginative and intriguing, but feel there is not enough experimental support for what is being proposed. The idea appears to be that after domain 4 binds substrate the catalytic domain is reeled into contact with the polysaccharide. This implies a substrate induced change in the conformational equilibrium but, as far as I can tell, the authors did not perform solution conformation studies in the absence AND presence of arabinan. Without the latter data there is an alternate explanation. In a most general sense, the primary role of a CBM is thought to be simply to maintain the proximity of CAZymes with a substrate thereby keeping high local substrate-associated concentrations of the enzyme. This requires productive collisions of the enzyme with the polysaccharide such that binding can occur. I find it possible that the conformational heterogeneity of AbnA may be a mechanism by which its CBM can simply sample more space to encourage the probability of binding-productive collisions with polysaccharide, and that the conformational changes are irrelevant to promoting direct contact of the catalytic module with substrate. I think the authors should be much more circumspect with their proposal of a mechanism unless they provide more data to support it.

The details of how the ITC was performed and analyzed is somewhat scant. It appears that the arabinan polymer in the syringe was titrated into a known concentration of protein in the sample cell. Reading between the lines, if I am correct, the analysis seems to have been done by using the standard "ligand in syringe" mode while manually adjusting the arabinan concentration (what this concentration was based on is also not stated in the manuscript) during fitting with a single site binding model until convergence with an n (stoichiometry) of 1. I acknowledge this approach has been used in the literature before but it is fraught with problems. At best, it gives OK results, at worst it can give outright wrong results. The first issue is a simple one where the human user who is manually adjusting the ligand concentration to get an $n=1$ becomes part of the minimization routine, which is not optimal. The second issue is that the chosen mode of fitting and the binding model assume the ligand in the syringe is monovalent (single class of binding site) and of known concentration, while the acceptor (in this case protein/enzyme) in the cell can be multivalent but also has a single class of binding site (these are simply features of the mathematical model). In the case of this study, the protein acceptor could be monovalent or divalent, with potentially non-equivalent binding to the catalytic site and CBM, while the polymer is definitely not monovalent (polysaccharides sometimes present multiple classes of non-equivalent binding sites but at worst can have much more complex scenarios). Notably, the actual concentration of binding sites in the polysaccharide preparation is unknown a priori. In short, the authors have forced the fitting scenario in a way it was never intended to be used, which can result in small to large errors in the regressed values, and some of the thermodynamic values do give me pause (e.g. differences between domains 123 and domains 12 defies a logical explanation). Only representative ITC experiments are given in the supplementary, but what I can say is that there does appear to be some systematic deviations in the fits. For example, the results with linear arabinan titrated into domain 4 actually looks like there may be multiple classes of binding sites. Assuming domain 4 is monovalent, this would potentially be attributable to two or more classes of binding sites on the arabinan. So, I am not so confident in the results as they have been analyzed. The best approach to analyzing such data is quite simple and involves switching the fitting mode where the protein in the cell is treated as the ligand and the polysaccharide in the syringe is treated as the acceptor. This gets around some of the assumptions and compromises, and allows a real stoichiometry for the polysaccharide to be determined and better potential for detection of complex binding. The authors then only have to assume integer valency (i.e. 1 or 2) for the protein (though the domain 1234 construct may still pose a problem). In short, though I have no doubt that there is binding at the catalytic site and the CBM, I encourage the authors to explore a more robust data analysis regime so that the individual binding sites at the catalytic site and CBM are more confidently and accurately demonstrated.

Following the NMA and MD simulations, the authors essentially jumped straight to model fitting with the SAXS to make an argument regarding flexibility of the protein. The observation that the multi-state models gave better fits to the SAXS data, thereby appearing to agree with the NMA/MD results, is a bit circular in its way, given that adding in more variables/models will almost always improve SAXS fitting, even in the absence of profound flexibility. I think the authors could make a more compelling case with a more direct and unbiased analysis of the SAXS data to examine protein flexibility. For insight into this, I refer the authors to Rambo and Tainer (2011) Characterizing flexible and intrinsically unstructured biological macromolecules by SAS using the Porod-Debye law. *Biopolymers*. 95(8):559-71. Such direct analyses of the scattering data should independently reveal flexibility thus additionally justifying the use of an ensemble approach to modelling the flexibility, beyond just revealing that ensemble models fit the data better.

Related to the above, throughout the manuscript the authors tend to refer to the some methods used as confirming one another. For example, on page the statement is made that the "...NMA results confirm the relevance of three observed crystallographic conformational states..." My feeling is to caution against uses of the word "confirm" as its connotation is the same as that of the word "proof," both of which imply absolute correctness, which is (virtually) impossible in the life sciences. NMA and MD are computational approaches and, as reliable as they may be considered, they are not real-world

observations of particular protein conformations. A similar argument can be applied to SAXS where one takes very low-resolution data and models it with "high resolution" pseudo-atomic models. I would be more comfortable if the authors took the approach that the results of one method "supports" the results of another, or "fails to disagree" with it, or some other language that is less absolute.

Finally, I found the glutathione locking approach really interesting and it provided some good results. However, it is very curious that the authors didn't kinetically characterize WT, oxidized E420C-W758C, AND reduced E420C-W758C. Without the later kinetics it was impossible to judge whether the lowered activity of the oxidized E420C-W758C was due to the locked conformation or the mutations themselves.

Reviewer #2 (Remarks to the Author):

The authors present several crystal structures of an arabinanase from the GH43 family, in which, for the first time, Domain 4 is also determined. A new binding site for arabinanase is found on Domain 4, with similar affinity to that of the catalytic site. By several computational, biochemical, and structural techniques the authors show that Domain 4 has large conformational flexibility relative to the other domains.

The arabinanase binding site on Domain 4 is completely unexpected, as far as I could judge from homology searches against the PDB. The flexibility of Domain 4 is also well established in this work. The authors suggest possible functional implications for these findings in the Discussion. It is likely that these new findings may be general in arabinanases, and perhaps in other enzymes for polysaccharide degradation.

I recommend publishing this manuscript with very minor revisions.

My comments are:

- 1) Figure 2C will be clearer if the caption states that the two conformations were superimposed based on Domain 4.
- 2) In pg 10, the three dihedral angles of the conformations are listed as: 8.0°, 8.7°, and 7.0°. Given the apparent angular change shown in Figure 2C, can the authors check that value of 7.0° is indeed correct (it is so similar to 8.0, and 8.7)?
- 3) The Metadynamics analysis appears to be very inaccurate. Primarily, because it puts two crystals structures (conf2 and conf3) in regions with extremely high free energies (>14kCal/Mol). It is hard to believe that crystal structures could exist in regions that are so prohibitive (even if they are at the edge of the map). The Metadynamics are also not very consistent with the standard MD, which favors a close conformation. The authors should put two additional stars on the map - corresponding to conf2 and conf3 - so that the readers could judge for themselves how reliable this analysis is.
- 4) I do not see the merit of Figure 7. It is just a recounting of all the other figures in the manuscript.

Rebuttal Letter- Manuscript COMMSBIO-19-1967A

Referee Information

Referee expertise:

Referee #1: Carbohydrate-binding modules

Referee #2: Structural biology

Reviewers' comments:

Reviewer #1 (Remarks to the Author):

This article describes primarily the structural analysis of a multimodular GH43 arabinanase, AbnA. The main outcomes of note are the discovery of a new carbohydrate-binding module and a full analysis of the conformational transitions of the enzyme. This is a technically very sound study with appropriate interpretations that I very much enjoyed reading and reviewing. There are in my opinion, however, some issues that I believe the authors should address.

We thank the reviewer for the positive assessment of our work.

In general, the tone of the article comes across as if the approach the authors have used to examine the conformational heterogeneity of a CAZyme is novel. This may have not been intentional, but the near complete lack of acknowledgement of similar studies performed in the past would make this seem the case to the uninitiated. I do believe the authors have used a particularly thorough suite of approaches; however, the examination of solution conformations of CAZymes is not new. X-ray crystallography (XRC) in combination with solution methods, largely focusing on SAXS but also including NMR, DLS, etc., has been applied to multimodular CAZyme for >15 years. In fact, a recent (2017) methods paper by Czjzek and Ficko-Blean outlines approaches to combine XRC with SAXS to examine multimodular CAZymes. A fairly large, but not complete, group of articles focusing on solution conformations of CAZymes are cited in that methods paper. These two authors have been quite influential in this area. As far as I can tell, the present study has failed to draw upon or acknowledge a fairly rich history of efforts in this field (more on this below).

We took note of this comment, and added many new references (ref. 25-33) in the introduction and discussion to previous works investigating multimodular CAZymes and their solution conformations.

We agree with the referee that work has been done in the past to study similar solution conformations, especially through the integration of XRC and SAXS. However, we feel that this work, which integrates **eight** different methods together for the study of such conformational flexibility, can be set apart from the other studies as a more thorough and comprehensive approach to study such flexibility. We not only provide models of different conformational states, but also provide **energetic** and **functional** information regarding the relevance of these conformations. We believe that the more comprehensive “integrative” approach used in this study may be useful for the study of many other ultra-multimodular enzymes.

In keeping with the above comment, the authors' apparent failure to incorporate the approaches and outcomes of similar past studies in their manuscript also means their results are not placed in the greater context of CAZyme research, and so it is not clear to the reader what the general significance may be. Indeed, the article by Lammerts van Bueren et. al. (Structure, 2011, 19:640–651) reveals the overall structure of a GH13 enzyme and its conformational transitions to be eerily similar to those reported by the authors. In comparison to this and other previous CAZyme studies that utilize SAXS, the relatively uncharacterized *M. viridifaciens* GH33 example that the authors use as a comparator seems very poor.

In light with this comment, we changed the introduction completely, and added changes to the discussion, focusing on the study of AbnA within the context of CAZyme research, providing many references to past studies, including comparisons to previous works such as the work by Lammerts van Bueren et al.

So, to summarize the previous two points, I believe the authors of this study have missed an opportunity to leverage a reasonably large body of existing literature in their study. To do so would not only informationally enriched their manuscript by providing more context but it would also help illuminate how the present approach

may build upon past efforts. This would help elevate the interest because, at present, a skeptic could view the manuscript as it is presented as simply a very detailed study of a single esoteric enzyme. This reviewer would like to think there is more to it than this.

We believe that the changes we made now to the introduction and discussion, detailed above, improve the manuscript in this respect.

On the basis of the conformations adopted by AbnA the authors propose a “harpoon” mechanism. I find this imaginative and intriguing, but feel there is not enough experimental support for what is being proposed. The idea appears to be that after domain 4 binds substrate the catalytic domain is reeled into contact with the polysaccharide. This implies a substrate induced change in the conformational equilibrium but, as far as I can tell, the authors did not perform solution conformation studies in the absence AND presence of arabinan. Without the latter data there is an alternate explanation. In a most general sense, the primary role of a CBM is thought to be simply to maintain the proximity of CAZymes with a substrate thereby keeping high local substrate-associated concentrations of the enzyme. This requires productive collisions of the enzyme with the polysaccharide such that binding can occur. I find it possible that the conformational heterogeneity of AbnA may be a mechanism by which its CBM can simply sample more space to encourage the probability of binding-productive collisions with polysaccharide, and that the conformational changes are irrelevant to promoting direct contact of the catalytic module with substrate. I think the authors should be much more circumspect with their proposal of a mechanism unless they provide more data to support it.

We took note of the referee’s comment, and modified this section so that the “harpoon” mechanism is presented in a more hypothetical light, while an alternative, more conservative mechanism is provided.

The details of how the ITC was performed and analyzed is somewhat scant. It appears that the arabinan polymer in the syringe was titrated into a known concentration of protein in the sample cell. Reading between the lines, if I am correct, the analysis seems to have been done by using the standard “ligand in syringe” mode while manually adjusting the arabinan concentration (what this concentration was based on is also not stated in the manuscript) during fitting with a single site binding model until convergence with an n (stoichiometry) of 1. I acknowledge this approach has been used in the literature before but it is fraught with problems. At best, it gives OK results, at worst it can give outright wrong results.

The first issue is a simple one where the human user who is manually adjusting the ligand concentration to get an $n=1$ becomes part of the minimization routine, which is not optimal. The second issue is that the chosen mode of fitting and the binding model assume the ligand in the syringe is monovalent (single class of binding site) and of known concentration, while the acceptor (in this case protein/enzyme) in the cell can be multivalent but also has a single class of binding site (these are simply features of the mathematical model). In the case of this study, the protein acceptor could be monovalent or divalent, with potentially non-equivalent binding to the catalytic site and CBM, while the polymer is definitely not monovalent (polysaccharides sometimes present multiple classes of non-equivalent binding sites but at worst can have much more complex scenarios). Notably, the actual concentration of binding sites in the polysaccharide preparation is unknown a priori. In short, the authors have forced the fitting scenario in a way it was never intended to be used, which can result in small to large errors in the regressed values, and some of the thermodynamic values do give me pause (e.g. differences between domains 123 and domains 12 defies a logical explanation). Only representative ITC experiments are given in the supplementary, but what I can say is that there does appear to be some systematic deviations in the fits. For example, the results with linear arabinan titrated into domain 4 actually looks like there may be multiple classes of binding sites. Assuming domain 4 is monovalent, this would potentially be attributable to two or more classes of binding sites on the arabinan.

So, I am not so confident in the results as they have been analyzed. The best approach to analyzing such data is quite simple and involves switching the fitting mode where the protein in the cell is treated as the ligand and the polysaccharide in the syringe is treated as the acceptor. This gets around some of the assumptions and compromises, and allows a real stoichiometry for the polysaccharide to be determined and better potential for detection of complex binding. The authors then only have to assume integer valency (i.e. 1 or 2) for the protein (though the domain 1234 construct may still pose a problem). In short, though I have no doubt that there is binding at the catalytic site and the CBM, I encourage the authors to explore a more robust data analysis regime so that the individual binding sites at the catalytic site and CBM are more confidently and accurately demonstrated.

We added in the experimental section the information of how the binding site concentration of arabinan was calculated. As for switching the fitting mode, this will require bringing the protein concentration in the syringe to close to 100 mg per ml, which is not feasible.

Following the NMA and MD simulations, the authors essentially jumped straight to model fitting with the SAXS to make an argument regarding flexibility of the protein. The observation that the multi-state models gave better fits to the SAXS data, thereby appearing to agree with the NMA/MD results, is a bit circular in its way, given that adding in more variables/models will almost always improve SAXS fitting, even in the absence of profound flexibility.

I think the authors could make a more compelling case with a more direct and unbiased analysis of the SAXS data to examine protein flexibility. For insight into this, I refer the authors to Rambo and Tainer (2011) Characterizing flexible and intrinsically unstructured biological macromolecules by SAS using the Porod-Debye law. *Biopolymers*. 95(8):559-71. Such direct analyses of the scattering data should independently reveal flexibility thus additionally justifying the use of an ensemble approach to modelling the flexibility, beyond just revealing that ensemble models fit the data better.

Following this comment, we provide besides the WT data, additional SAXS data collected on truncation mutants of AbnA (AbnA-D123 and AbnA-D12), and compare all three data with the Prod-Debye law to demonstrate significant flexibility for the full-length enzyme (**Supplementary Fig. 4, revised SAXS results section**).

We also provide the SAXS envelopes calculated from these data, to provide a qualitative fit to the structural data, as demonstrated similarly in previous works and specifically, in the (2017) methods paper by Czjzek and Ficko-Blean.

However, we believe that our method of multi-state modeling to fit the SAXS profiles is a more quantitative and reliable method to compare structural data to SAXS data than fitting structures to SAXS envelopes, and provides more information. From experience, we do not believe that when providing any 3 models to fit the SAXS profiles, the SAXS fit will always improve; most times it does not improve at all, and we believe that it only improves significantly when the different models are correct (obviously within a certain error). Of course, if we would have provided >10 models to fit the SAXS data, the fit may have improved regardless of the structures, however, we do not believe this is the case when providing only 3 models. A similar approach to fit the SAXS data through multi-state modeling can be found in Schneidman-Duhovny et al (2016) *Nucleic Acids Research* **44**:W424-W429.

Related to the above, throughout the manuscript the authors tend to refer to the some methods used as confirming one another. For example, on page the statement is made that the "...NMA results confirm the relevance of three observed crystallographic conformational states..." My feeling is to caution against uses of the word "confirm" as its connotation is the same as that of the word "proof," both of which imply absolute correctness, which is (virtually) impossible in the life sciences. NMA and MD are computational approaches and, as reliable as they may be considered, they are not real-world observations of particular protein conformations. A similar argument can be applied to SAXS where one takes very low-resolution data and models it with "high resolution" pseudo-atomic models. I would be more comfortable if the authors took the approach that the results of one method "supports" the results of another, or "fails to disagree" with it, or some other language that is less absolute.

We took note of the referee's comment, and changed the language accordingly throughout the text.

Finally, I found the glutathione locking approach really interesting and it provided some good results. However, it is very curious that the authors didn't kinetically characterize WT, oxidized E420C-W758C, AND reduced E420C-W758C. Without the later kinetics it was impossible to judge whether the lowered activity of the oxidized E420C-W758C was due to the locked conformation or the mutations themselves.

We agree with the reviewer. We failed to mention in the text that the AbnA (E420C, W758C) mutant has essentially the same activity as of the WT enzyme. Only under oxidizing conditions (in the presence of 5mM GSSG) the mutant losses its activity. The information is provided now in the results section.

Reviewer #2 (Remarks to the Author):

The authors present several crystal structures of an arabinanase from the GH43 family, in which, for the first time, Domain 4 is also determined. A new binding site for arabinanase is found on Domain 4, with similar affinity to that of the catalytic site. By several computational, biochemical, and structural techniques the authors show that Domain 4 has large conformational flexibility relative to the other domains.

The arabinanase binding site on Domain 4 is completely unexpected, as far as I could judge from homology searches against the PDB. The flexibility of Domain 4 is also well established in this work. The authors suggest possible functional implications for these findings in the Discussion. It is likely that these new findings may be general in arabinanases, and perhaps in other enzymes for polysaccharide degradation.

I recommend publishing this manuscript with very minor revisions.

My comments are:

1) Figure 2C will be clearer if the caption states that the two conformations were superimposed based on Domain 4.

Done.

2) In pg 10, the three dihedral angles of the conformations are listed as: 8.0°, 8.7°, and 7.0°. Given the apparent angular change shown in Figure 2C, can the authors check that value of 7.0° is indeed correct (it is so similar to 8.0, and 8.7)?

This was checked, and is correct. The reason for the apparent small changes is that this angle is the **dihedral** angle of all domains relative to one another, which is almost the same when comparing Conf2 and Conf3. The relatively large angle between the states that is shown in Figure 2C is in consequence to an **internal rotation** of ~45° of Domain4 around itself.

To clarify this point, the text was changed accordingly (page 6, lines 7-8), and an additional panel of comparison between the two structures, viewed from the side, was added to Figure 2.

3) The Metadynamics analysis appears to be very inaccurate. Primarily, because it puts two crystal structures (conf2 and conf3) in regions with extremely high free energies (>14kCal/Mol). It is hard to believe that crystal structures could exist in regions that are so prohibitive (even if they are at the edge of the map). The Metadynamics are also not very consistent with the standard MD, which favors a close conformation. The authors should put two additional stars on the map - corresponding to conf2 and conf3 - so that the readers could judge for themselves how reliable this analysis is.

True, the other two crystal structures are located at the periphery of the map, in a high energy region. However, we believe that the error of these positions is within the error of the method, especially since we had to introduce upper and lower restraining potential walls into the CV1 collective variable to assist the simulation in reaching convergence, preventing thus much sampling of “very” closed conformations (see methods). Although this obviously introduces some artifact into the landscape, we believe that qualitatively, this energy map is a relatively good representative of the real energy map for AbnA, which is obviously much more complex. Thus, we got low energy minima for relatively “closed” conformations, and higher energy local minima for “wide-open” conformations. Such results **suit** the standard MD simulations, where the “closed” states are sampled more frequently owing to a lower energy state. Thus, although the other two crystal structures are in the periphery of the map, they are relatively close (within the error of the method) to the EM2 local minima. Within the error of the method, and in light of the significantly larger conformational changes the enzyme could undergo, all three crystal structures could be considered as a single “closed” conformation, corresponding generally to EM2.

In any case, however, additional stars were added to indicate the positions of the other two crystal structure conformations.

4) I do not see the merit of Figure 7. It is just a recounting of all the other figures in the manuscript.

After some consideration, we prefer to leave Figure 7 in the main text. We think that Figure 7 does a good job in summarizing the methodology used in this work, and in presenting in a clear way the steps used in the “integrative structure determination” approach. Since we think this paper could be of *methodological* use to other researchers studying ultra-multimodular proteins, we think this figure is very important and central to understanding more clearly the workflow that was used and that can be used in other cases researching ultra-multimodular proteins.

** See Nature Research's author and referees' website at www.nature.com/authors for information about policies, services and author benefits

COMMSBIO - This email has been sent through the Springer Nature Tracking System NY-610A-NPG&MTS

REVIEWERS' COMMENTS:

Reviewer #1 (Remarks to the Author):

The manuscript by Lansky et al describes the multi-pronged approach to the conformational and functional analysis of a multimodular GH43 arabinanase, AbnA. As with the previous version this was a very interesting read and this current version is much improved over the last, now with a more compelling narrative. In general, the authors have done a very nice job of dealing with my criticisms. The remaining issues result from the changes made in this iteration, but are very minor.

1. On page 8, line 5, when the authors mention the Porod-Debye analysis they correctly interpret it to reveal flexibility in the full-length AbnA. However, in the parentheses they imply this is because of the presence of a plateau in the plot, where in fact it is the opposite – the lack of a plateau means flexibility and the AbnA plot shows NO plateau. Regarding this, my request for this analysis was to demonstrate flexibility and justify its incorporation into SAXS modelling, but I did not ask for the generation of molecular envelopes (which can get messy with flexible proteins). Nevertheless, their inclusion does not weaken the arguments, and in some ways enhances them.

2. The inclusion of the unitary n values in the ITC table is pointless as they really don't mean anything. A better value would be to show this as the footprint, which would be the known concentration of the polysaccharide expressed as monosaccharide equivalents divided by the ligand concentration the user had to input to get an n of 1. This should be a value greater than 1 that meaningfully represents how many monosaccharide equivalents of polysaccharide chain is, on average, required for binding. Again, regarding this, in my previous review where I requested a reanalysis of the ITC I did not mean that the protein should be what was injected into the cell. I recognize this tends to require beyond what is reasonable for protein solubility. I meant that the software provides a means by which existing data can be reanalyzed by simply reversing what is considered ligand and what is considered the acceptor. It is a simple menu selection available in MicroCal Origin ITC analysis software. All of the data could be easily and more accurately reanalyzed by making this simple switch – no more experimentation, no requirement of the user to become part of the minimization routine, and a more direct fitting of the stoichiometry. However, I feel that reiterating this request at this point would be unreasonable and leave it to the authors discretion whether they want to do this or not.

Reviewer #2 (Remarks to the Author):

My concerns have been addressed.

I recommend that the manuscript is accepted for publications.

Rebuttal Letter- Manuscript COMMSBIO-19-1967A

Reviewers' comments:

REVIEWERS' COMMENTS:

Reviewer #1 (Remarks to the Author):

The manuscript by Lansky et al describes the multi-pronged approach to the conformational and functional analysis of a multimodular GH43 arabinanase, AbnA. As with the previous version this was a very interesting read and this current version is much improved over the last, now with a more compelling narrative. In general, the authors have done a very nice job of dealing with my criticisms. The remaining issues result from the changes made in this iteration, but are very minor.

1. On page 8, line 5, when the authors mention the Porod-Debye analysis they correctly interpret it to reveal flexibility in the full-length AbnA. However, in the parentheses they imply this is because of the presence of a plateau in the plot, where in fact it is the opposite – the lack of a plateau means flexibility and the AbnA plot shows NO plateau. Regarding this, my request for this analysis was to demonstrate flexibility and justify its incorporation into SAXS modelling, but I did not ask for the generation of molecular envelopes (which can get messy with flexible proteins). Nevertheless, their inclusion does not weaken the arguments, and in some ways enhances them.

Thank you for noticing this mistake. We changed the text accordingly (page 8 lines 5-7).

2. The inclusion of the unitary n values in the ITC table is pointless as they really don't mean anything. A better value would be to show this as the footprint, which would be the known concentration of the polysaccharide expressed as monosaccharide equivalents divided by the ligand concentration the user had to input to get an n of 1. This should be a value greater than 1 that meaningfully represents how many monosaccharide equivalents of polysaccharide chain is, on average, required for binding. Again, regarding this, in my previous review where I requested a reanalysis of the ITC I did not mean that the protein should be what was injected into the cell. I recognize this tends to require beyond what is reasonable for protein solubility. I meant that the software provides a means by which existing data can be reanalyzed by simply reversing what is considered ligand and what is considered the acceptor. It is a simple menu selection available in MicroCal Origin ITC analysis software. All of the data could be easily and more accurately reanalyzed by making this simple switch – no more experimentation, no requirement of the user to become part of the minimization routine, and a more direct fitting of the stoichiometry. However, I feel that reiterating this request at this point would be unreasonable and leave it to the authors discretion whether they want to do this or not.

We agree with the reviewer and modified the ITC Table accordingly.

Reviewer #2 (Remarks to the Author):

My concerns have been addressed.

I recommend that the manuscript is accepted for publications.